# GIFARC: SYNTHETIC DATASET FOR LEVERAGING HUMAN-INTUITIVE ANALOGIES TO ELEVATE AI REASONING

## ABSTRACT

The Abstraction and Reasoning Corpus (ARC) poses a stringent test of general AI capabilities, requiring solvers to infer abstract patterns from only a handful of examples. Despite substantial progress in deep learning, state-of-the-art models still achieve accuracy rates of merely 40-55% on the 2024 ARC Competition, indicative of a significant gap between their performance and human-level reasoning. In this work, we aim to bridge that gap by introducing an analogy-inspired dataset called GIFARC. Leveraging vision-language models (VLMs), we synthesize new ARC-style tasks from a variety of GIF images that include analogies. Each new task is paired with a ground-truth analogy, providing an explicit mapping between visual transformations and everyday concepts. By embedding robust human-intuitive analogies into ARC-style tasks, GIFARC guides AI agents to adopt analogical reasoning approaches, facilitating more concise and human-understandable solutions. We empirically demonstrate that GIFARC improves task-solving performance by aligning model reasoning with human analogical problem-solving strategies.[1][2]

## 1 INTRODUCTION

Despite recent advances in AI, many of current AI models remain limited in their capacity for flexible, general-purpose reasoning, which is a core characteristic often associated with artificial general intelligence (AGI). In this context, the Abstraction and Reasoning Corpus (ARC) was introduced as a benchmark aiming to measure the ability of artificial general intelligence systems to abstract visual patterns and reason about them in a manner akin to human cognition (Chollet, 2019). This requirement closely mirrors the cognitive flexibility inherent to human task-solving, thus making ARC an essential playground for probing the fundamental limitations and capabilities of current AI systems in their pursuit of human-level AGI.

In the 2024 ARC Prize competition and semi-evaluation, various deep neural networks achieved strong performance, such as J.Berman (Berman, 2025) and E.Pang (Pang, 2025) reached 79.6% and 77.1% respectively. However, most models saw their performance reduced by more than half on ARC-AGI-2, which lacks data generators like ReARC (Hodel, 2024) for training on similar tasks. This reveals a fundamental limitation: while deep neural networks excel with ample data, they struggle in example-deficient tasks requiring abstract and compositional reasoning (Dziri et al., 2023; Lee et al., 2024). ARC thus provides strong evidence that data-driven approaches alone do not yet capture the breadth of human reasoning and generalization.

Why does this limitation exist? One crucial factor lies in how humans leverage analogies to map new visual or conceptual clues onto existing knowledge. When people see a grid transformation as in Figure 1, they can instantly map this visual pattern onto familiar concepts such as gravity and color change. These priors, accumulated from a lifetime of diverse visual and conceptual experiences, provide immediate insight into the task constraints and sharply reduce the complexity of the tasks. Rather than searching blindly through every possible transformation, humans can reject lengthy or counterintuitive candidates and focus on the small subset of patterns consistent with their analogies.

---

[1] Tasks generated with GIFARC are visualized in `https://gifarc.vercel.app/`.
[2] Full dataset is available at `https://huggingface.co/datasets/DumDev/gif_arc`.

Figure 1: Illustration of two different solutions of ARC-style task found with or without analogic approach. In general, humans will perceive the solution of (b) better than (a), as it is more concise and easier to understand. When the agent has been guided to think analogically with GIFARC dataset, as depicted in (b), it seeks a more concise and human-intuitive solution.

In contrast, deep learning models rarely have an explicit mechanism for analogy-making (Mitchell, 2021). Even advanced neural networks typically rely on statistical correlations learned from large training sets, lacking the means to "tag" or "reframe" new tasks in terms of previously understood scenarios (Bober-Irizar & Banerjee, 2024). Consequently, these models miss a powerful heuristic: an analogy-driven search strategy that narrows down the solution space by linking visual abstractions to real-world events.

**Need of analogy-contained dataset.** To help bridge this gap between human cognition and AI, the agent should obtain human-level amount of analogies in prior to any task-specific training. Since the models we aim to train are predominantly data-driven, the need for training dataset that consists of useful analogies naturally arise. The dataset we seek contains following criteria;

(i) *ARC-style examples*: the dataset should contain analogies in forms of ARC-style examples. Since the agent is evaluated via ARC benchmark, the agent has the world perception in form of 2D grids. In addition, the analogies one needs to know should be expressible on an ARC-style 2D grid. Thus, the analogy we intend to train should also be expressed and given in the 2D grid in order for the agents to understand and utilize them during evaluation.

(ii) *Executable codes*: For the agent to capture how to manipulate 2D grids in accordance with the analogy, the transformation should be provided as lines of executable code. With these 2D grids and codes, the agent should be able to correlate ARC-style transformations with their underlying analogies and identify them during evaluation.

In this paper, we propose an analogy-inspired synthetic ARC dataset, GIFARC. Drawing on vision-language model (VLM), we developed a framework to synthesize ARC-style tasks from online GIF images by extracting various visual patterns and the analogies they evoke. These GIFs encompass a range of scenarios (e.g., snowfall, blooming flowers, moving objects), each providing a different type of visual transformation. We encode these transformations in newly generated ARC-style tasks and, crucially, supply ground-truth analogy labels that clarify the underlying conceptual mapping (e.g., "water flow blocked by an obstacle").

**Role of GIF as the basis of ARC-style task generation.** Using GIFs as the starting ingredient of ARC-style tasks, our GIFARC synthesis pipeline takes great advantages in picking human-understandable and visualizable analogies, as GIFs are series of images that are intentionally drawn to visualize certain ideas. In addition, GIF also has advantages in holding a sufficiently small number of analogies. In terms of analogy extraction simplicity, it is critical for the analogies to be implied in the image series without any entanglement to one another. Longer image series such as videos are more likely to hold excessive number of analogies and consequently have a higher risk of entanglement in between. Thus, GIF is an adequate form of analogy-implying image series.

These new ARC-style tasks generated from GIFs will possess a variety of intuitive analogies. The agent trained with this dataset will be guided to gain the grasp of analogical approaches and utilize them in the solving step. By incorporating diverse analogies into the dataset, model trained with GIFARC is guided to "think analogically" before searching for solutions, thereby reducing the complexity of task-solving. In addition, by converting the visual data to verbose expression before generating the ARC-style tasks, the agent will specifically gain the ability to verbally point out the analogy intended in the ARC-style task.

Running this pipeline on motion-rich GIFs produces 10k unique high-quality ARC-style tasks spanning 20 analogy categories. We empirically validated GIFARC through two complementary approaches. First, in-context learning experiments demonstrated that GIFARC effectively guides LLMs toward human-like analogical reasoning, as models using full analogical descriptions achieved better alignment with ground-truth analogies than those using analogy-removed descriptions. Second, supervised fine-tuning demonstrated quantitative improvements, with accuracy increasing from 0.2% to 2.9% on ARC-AGI-1 and negative log-likelihood decreasing by 94.5% on ARC-AGI-2.

## 2 RELATED WORK

**Abstraction and Reasoning Corpus.**   The Abstraction and Reasoning Corpus (ARC), introduced by Chollet (2019), is an open-ended benchmark designed to evaluate broad generalization from minimal demonstrations. Its goal is to test how efficiently an agent can acquire and apply abstract patterns with just a few examples. In 2020, there have been many attempts to solve ARC through Domain Specific Language (DSL) based brute force programs or various search algorithms. The limitation of these methodologies was that they required manually written DSLs that humans deemed necessary. Since the release of GPT-4 in 2023, the performance of LLMs has been improved significantly, leading to various attempts to solve ARC problems using LLMs (Mirchandani et al., 2023; Xu et al., 2024; Lee et al., 2024). In 2024, there were two LLM utilized approaches. The first approach, similar to previous methods, involved LLMs directly predicting test outputs. ARCitect (Franzen et al., 2024) won first place in the 2024 ARC Prize by fine-tuning models using data augmented with ReARC (Hodel, 2024) and employing specialized decoding strategies for predicting test outputs. Additionally, OpenAI's reasoning models, such as o1 and o3, showed significant performance despite only applying CoT, unlike the approaches from 2023. The second approach using LLMs generated Python code solutions with LLMs (Li et al., 2024; Berman, 2024). Berman (2024) generated over 250 Python codes from LLMs for a single problem and refined them to create an optimal solution code. However, this approach has the disadvantage of requiring high costs due to the need to generate many code candidates. In 2025, a new version of the ARC benchmark named ARC-AGI-2 (Chollet et al., 2025) was released, which requires more abstract and analogical thinking compared to the existing ARC-AGI-1. Former ARC-AGI-1 winning models struggled to solve the new version benchmark, implying that the prior approaches still have room for development.

**Synthetic ARC Generation.**   To address the data scarcity inherent in ARC, which hinders both symbolic program synthesis and the training of neural models, researchers have turned to generating programmatic data using human-curated, AI-generated, or hybrid pipelines (LeGris et al., 2024; Opiełka et al., 2024; Qi et al., 2021). One such effort is ReARC, which manually scripts generative procedures for each task to produce an effectively infinite set of in-distribution examples (Hodel, 2024). LARC (Language-Complete ARC) crowd-sources natural language explanations of tasks, enabling models to learn from natural programs instead of raw grids (Acquaviva et al., 2022). Other benchmarks such as ConceptARC, Mini-ARC, and 1D-ARC use manually crafted rules to produce diverse yet interpretable task distributions (Kim et al., 2022; Moskvichev et al., 2023; Xu et al., 2024). Recent approaches automate this process at scale. For instance, BARC generates 400,000 training examples by prompting GPT-4 to mutate a curated set of 160 Python-based solvers (Li et al., 2024). Similarly, MC-LARC transforms ARC tasks into multiple-choice formats, enabling scalable evaluation of language models' intermediate reasoning stages (Shin et al., 2024).

Different from existing in-distribution resampling or code-mutation pipelines, our work introduces GIFARC, a new dataset that mines human-intuitive GIFs, distills their analogical structure with vision-language models, and auto-compiles each analogy into an executable ARC-domain specific language (ARC-DSL) code, explicitly teaching models how to think by analogy rather than simply giving them more examples to memorize.

## 3 METHOD

### 3.1 OVERVIEW

**Problem formulation.** Given a set of GIF images $\mathcal{G} = \{g_j\}_{j=1}^N$, our goal is to construct an analogy-grounded ARC dataset, GIFARC, containing ARC-style tasks and represented as triples $\mathcal{T} = (\mathcal{E}, \alpha, \phi)$ where (i) $\mathcal{E} = \{(x_i, y_i)\}_{i=1}^K$ is a set of $K$ input-output grid pairs of ARC-style task. Here, $x_i \in \{0, \ldots, 9\}^{h_{x,i} \times w_{x,i}}$ is the input grid, $y_i \in \{0, \ldots, 9\}^{h_{y,i} \times w_{y,i}}$ is the output grid, with $h_{G,i}, w_{G,i}$ respectively being the height and width of the $G$ grid (either grid $x$ or $y$) in $i$-th pair of $\mathcal{E}$; (ii) $\alpha$ is a short natural-language analogy description (e.g., "blocked water flow"). (iii) $\phi$ is a Python program implementing $\mathcal{F} : \mathcal{X} \to \mathcal{Y}$, the latent deterministic transformation such that $y_i = \mathcal{F}(x_i)$; GIFARC ($\mathcal{D} = \{\mathcal{T}_j\}_{j=1}^N$) therefore pairs each ARC task with its provenance and analogy.

**Pipeline summary.** Our data synthesis pipeline is a three-stage process that transforms raw GIFs into fully executable, analogy-grounded ARC-style tasks (see Figure 2 for illustrative examples).

$$\text{GIF} \to \underbrace{\text{Visual Abstraction}}_{\S\,3.2} \to \underbrace{\text{Task Sketch}}_{\S\,3.3} \to \underbrace{\text{Executable ARC Task}}_{\S\,3.4}$$

Step 1. A VLM (o4-mini) analyzes each GIF $g$ and outputs a structured JSON record that captures its objects, static/dynamic patterns, interactions, and core reasoning principles $\mathcal{A}(g)$.

Step 2. A VLM (o4-mini) converts the JSON abstraction into a concise ARC-style task sketch: a set of concepts with a natural-language description that implies the extracted patterns.

Step 3. Conditioned on the sketch and a handful of retrieved ARC task examples, the VLM (o4-mini) synthesizes Python code implementing `generate_input` (stochastic examples) and `main` (deterministic transformation). The output is a complete ARC-style task comprising (i) an input-output demonstration set $\mathcal{E}$, (ii) the Python solution program $\phi$, and (iii) its analogy label $\alpha$.

Details of Step 1 appear in Section 3.2, Step 2 in Section 3.3, and Step 3 in Section 3.4. All model calls use Azure OpenAI endpoints. The GIFARC dataset generated by this pipeline is analyzed in Appendix A.

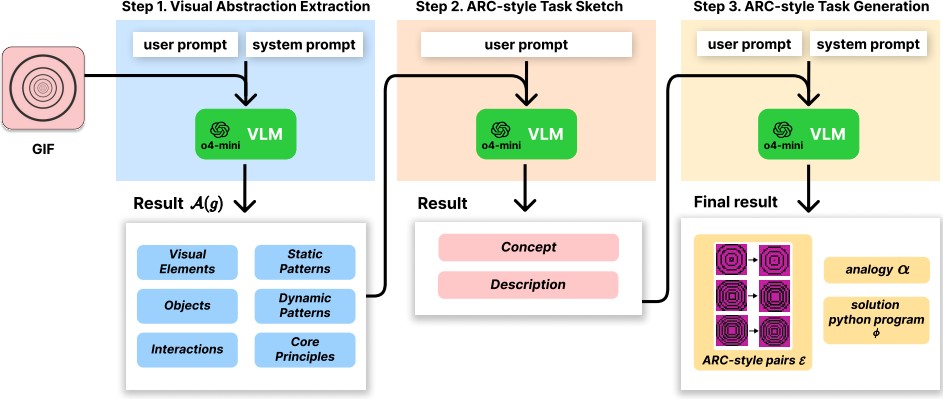

Figure 2: Illustration of GIFARC data synthesis pipeline that transforms a single GIF into a corresponding ARC-style task and supplementary data. In step 1, o4-mini digests a GIF file and outputs a detailed text expression $\mathcal{A}(g)$ of the visual transformation implied in the GIF in a JSONL format. In step 2, o4-mini reads step 1 result JSONL and outputs a text sketch an ARC-style task. In step 3, o4-mini reads the step 2 sketch result and generates an ARC-style task $\mathcal{E}$, an implied analogy $\alpha$, and the Python solution program $\phi$. The example prompts used in each step are reported in Appendix B.

### 3.2 STEP 1. EXTRACTING VISUAL ABSTRACTIONS FROM GIFs

Our first stage converts a raw GIF into a structured, readable summary that captures the analogy-laden visual logic hidden in the clip. Given an animated GIF $g$ depicting a short visual phenomenon (e.g.,

snow piling up, light reflecting, pipes clogging), we wish to automatically extract the key elements $\mathcal{A}(g)$ that a human would use when reasoning by analogy: what the scene contains, which parts move or remain fixed, how objects interact, and which general principle explains the observed dynamics. The output, called a visual abstraction, serves as the input for the later stage.

**GIF crawl.** To maximize coverage of distinct visual regularities, we first assemble a visual transformation corpus of GIFs. Using the GIPHY API, we query a curated tag list {nature, geometry, pattern, relational_pattern, sequence} where these types of clips often exhibit dynamic change. In particular, we crawled those with G-rated ratings, to generate nonharmful ARC-style tasks and prevent possible refusal from the following LLM APIs throughout the pipeline. After removing duplicates, we obtained 10k GIFs in total.

**Prompt for extracting visual abstractions.** We then submit every GIF to the vision-language model o4-mini under a prompt intentionally designed around two objectives: (i) *granularity* - to extract as many diverse visual analogies as possible from the GIF, (ii) *structure* - the reply must be valid JSON, and (ii) *coherence* - to ensure that each extracted visual analogy maintains alignment with the original GIF. Full prompt used in this step is reported in Appendix B.1. The system message states the structural constraint ("return JSON with keys {visual elements ... core principles} and use bullet lists only"). The user message enumerates the six fields and attaches concrete bullet-level instructions.

These six fields in $\mathcal{A}(g)$ serve complementary purposes: (i) visual_elements stores the raw, static pixel facts; (ii) objects binds memorable names to entities so later prompts can reference them; (iii) interactions records explicit or implicit relations (e.g., "snowflake collides with ground", "fluid flow is blocked") among objects. (iv) static_patterns and (v) dynamic_patterns distinguish invariants and variants from transformations, respectively; (vi) core_principles captures the very abstraction (e.g., gravity, reflection, occlusion, ...) that we ultimately hope an ARC solver will recall as an analogy.

This step is an extraction step to pull out intuitive analogies from GIF image. Converting pixel data into a more lightweight text descriptions lets the pipeline to efficiently digest the analogy faster through the remaining synthesis stages without having to handle heavy image data along. Also, in case that GIF holds multiple visual analogies, explicitly pointing out which visual analogy to focus on lets the agents in later stages interpret the target analogy without possible confusion. Furthermore, this stage is critical in abiding the GIPHY API terms of right. By irreversiblly conceptualizing visual data into text description, we have generated a distinct visual analogy data pool that does not contain any explicit GIFs.

## 3.3 STEP 2. FROM VISUAL ABSTRACTION TO TASK SKETCH

The visual abstraction $\mathcal{A}(g)$ obtained in previous step is rich but verbose; it contains a lot of observable details of the GIF, many of which may be irrelevant for task design (e.g., background textures, camera shakes). Before we can synthesize executable code, we therefore distill $\mathcal{A}(g)$ into a compact *task sketch* composed of two parts: (i) **Concepts:** 3-5 keyword phrases that name the core visual ideas (e.g., blocked_flow, rotational_symmetry, incremental_accumulation, ...); (ii) **Description:** a single paragraph that narrates how those concepts manifest on an ARC-style grid.

**Prompt for task sketch.** We obtain each sketch from a single completion of the vision language model o4-mini using a three-block prompt: **Block 1** provides in-context demonstrations; we draw 75 concepts/description pairs at random from the 160 human-written example tasks released with the BARC corpus (Li et al., 2024) and paste them. Because these examples cover symmetry, counting, physical analogy, topological change, and more, they implicitly teach the model what "looks like" a solvable ARC-style task while leaving room for new content. **Block 2** provides the six fields of $\mathcal{A}(g)$ from the previous step in bullet form. **Block 3** has a message that freezes the output format, exactly two comment headers for concepts and description.

We further instruct the model to reflect the full trajectory from initial to final state, maintaining semantic fidelity between the GIF dynamic behavior and the input-output structure of ARC-style grid. Full prompt used in this step is reported in Appendix B.2.

This selection step is crucial in two aspects. First, we expect VLM to filter out abstractions that cannot be turned into deterministic grid transformations (e.g., 'clouds forming random shapes'). Second, it forces the retained content into the exact format expected by our code-generation prompt via in-context learning, thereby lowering entropy and reducing failure cases downstream.

### 3.4 STEP 3. GENERATING EXECUTABLE ARC-STYLE TASKS

The final stage transforms each task sketch produced in Step 2 into a fully executable ARC-style task. The output comprises two parts: (i) `main`, a deterministic Python function implementing the latent transformation $\mathcal{F}$ (rendered as Python code $\phi$); (ii) `generate_input`, a stochastic Python function that draws fresh grid examples from the same distribution to generate input grids.

From this output, input grids generated with `generate_input` function are fed to the `main` function to produce corresponding output grids, thereby constructing a set of input-output grid pairs $\mathcal{E}$. The analogy $\alpha$ is copied from the task sketch created in Step 2, forming a ARC-style task $\mathcal{T} = (\mathcal{E}, \alpha, \phi)$.

**Prompt for generating executable ARC-style tasks.** Our code-generation prompt inherits the overall structure introduced in BARC (Li et al., 2024), but differs in that the new task is grounded in the GIF-derived concepts and description produced in Step 2. To tighten this grounding while preserving the coding style that has been proven effective in earlier literature, we again adopt an in-context learning strategy, along with retrieval augmentation. Specifically, we embed both our description and the 160 human-authored example ARC-style tasks with `text-embedding-ada-002` model. The cosine similarity between sketch and examples is computed, and the top-4 semantically closest examples are pasted into the prompt before the new sketch. Each example already contains a working `generate_input` / `main` pair, so these few-shot examples show the model how high-level concepts and narrative prose translate into concise Python code. The full prompt example for generating executable ARC tasks is reported in Appendix B.3.

## 4 APPLICATIONS

In this section, we conducted three experiments to examine: (1) how in-context learning with appropriately selected guiding examples influences the task-solving strategy of a LLM; (2) how this assists in identifying ground-truth analogies; (3) how it enhances the model performance in ARC-AGI evaluation tasks.

### 4.1 VERIFICATION CHANGES IN REASONING STEPS

Is the components in GIFARC really meaningful in analogical understanding of ARC-AGI tasks? In order to check the importance of each element in the dataset, we 'flattened' some key parts in GIFARC dataset sample and tested if the presence of components in GIFARC show significant difference in model understanding of the task.

We prepared three types of processed GIFARC data subset with different levels of components included:

(i) **full description**: 15 well-figuratively described data $(\mathcal{E}, \alpha, \phi)$ chosen from GIFARC through iterative refinement with LLM and human.

(ii) **description without analogy**: 15 data $(\mathcal{E}, \alpha_{\text{flat}}, \phi_{\text{flat}})$ from full description but gone through a 'flattening' process where researchers replaced analogic terms in $\alpha$ and $\phi$ into their corresponding lower-level synonyms.

(iii) **description without analogy and without solution**: 15 data $(\mathcal{E}, \alpha_{\text{flat}})$ from description without analogy but without solution $\phi_{\text{flat}}$.

Note that smaller numbered description contains richer analogies. These in-context learned models were then prompted to solve the tasks of ARC-AGI-2. In order to effectively check whether an analogic approach was used, models were instructed to specify two points in the result; first identify the underlying analogy in the given task, and implement it as a detailed solution.

For each different sampled GIFARC data as a context, GPT 4.1-mini was instructed to return the thought process while solving ARC-AGI-2 public evaluation dataset. As described above, since ARC-AGI-2 dataset requires a more diverse approach than the predecessor, analogic approach in the task-solving process would be more clearly shown and used for the comparison of the reasoning step.

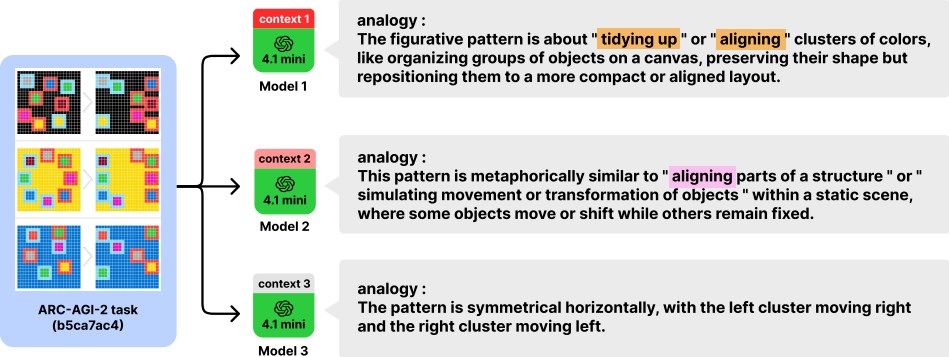

Figure 3: Experiment 1 case study: the LLMs learned in-context with richer analogy context result in more analogic interpretation of the ARC-AGI-2 task. Context 1, 2, 3 respectively stands for full description sample, description sample without analogy, description sample without analogy and without solution.

As a result of the solution comparison between models, we could confirm instances that analogies in GIFARC significantly helped identify the analogy of ARC-AGI-2 task. GPT 4.1-mini with full description expressed the analogy applied to the ARC-AGI-2 problem using an intuitive analogy. Figure 3 depicts one of the well-performed result cases. While GPT 4.1-mini with description without analogy and without solution mainly described the task analogy with grid-level terms (symmetrical, horizontal, left, right), GPT 4.1-mini with full description was able to describe with words from several domains (such as 'tidying up' and 'organizing'). In particular, it was observed that GPT 4.1-mini perceived full description as not only as an 'analogy phrase pool', but also recognized it as an 'example of analogic approach' and could use external domain terminologies those are not present in context. This result shows that the analogic terms that were wiped during the flattening process are critical for the LLMs to get the grasp of analogic approach needed in solving ARC-style tasks.

## 4.2 ALIGNMENT BETWEEN CONTEXT-GENERATED AND GROUND-TRUTH ANALOGIES

We have now confirmed that the components are analogically meaningful. Then are those analogies understandable and aligned to human perception? This experiment was conducted to verify if LLMs with GIFARC-sampled context can successfully identify underlying analogy in ARC-style tasks.

Here, we used full description and analogy-removed description, which we further removed the analogical expressions that remained in description (such as "gravity", "spiral spin", and "flicker") and replaced them with grid-level descriptions (such as "move down", "rotate", and "color change"). Next, GPT 4.1-mini with full description and GPT 4.1-mini with analogy-removed description were instructed to find the analogy implied in the 12 ARC-style tasks. Along with two models, three ARC expert humans also collected analogies implied in the 12 tasks. The outputs of two models and three humans were then compared with the ground-truth analogies of the GIFARC task.

We first instructed o3-mini to evaluate semantic similarities by providing a one-shot guideline about measuring similarity. This evaluator LLM evaluated the semantic similarities between analogies generated by solvers (two in-context learned LLMs and three humans) and the ground truth analogy.

Figure 4a depicts the similarity result done by LLM. It resulted that the output of GPT 4.1-mini with full description showed 0.138 similarity, while the output of GPT 4.1 mini with analogy-removed description only measured 0.050. In order to cross-check the reliability of the evaluation done by LLM, we re-conducted the experiment using human evaluators instead of an LLM-based evaluator. Five human evaluators ranked the analogy answers of five agents (three humans and two models) for 12 GIFARC tasks, ordering them based on their similarity to the original GIFARC analogies. As a result, the analogy of three human agents ranked on 1st place for all evaluators, the model

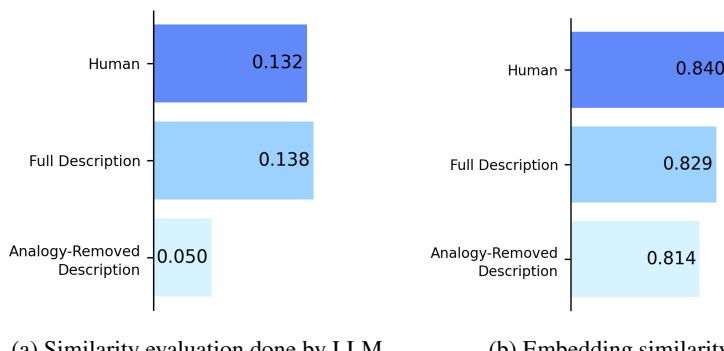

(a) Similarity evaluation done by LLM        (b) Embedding similarity

Figure 4: Similarity analysis between task-implied analogy found by GIFARC-trained LLMs and its corresponding hand-crafted analogy identified by three ARC experts. Figure 4a illustrates that the analogy generated by GPT 4.1 mini with full description showed better similarity than that of GPT 4.1 mini with analogy-removed description by 8.7 pp, and embedded cosine similarity by 1.4 pp, as depicted in Figure 4b.

trained with full analogies through in-context learning ranked at 2.33, and the model trained with analogy-removed descriptions ranked at 2.5 on average. These results showed consistency with the rankings obtained using the LLM evaluator in the original experiment.

Secondly, we embedded each outputs and compared the cosine similarity between. It resulted that the output of GPT 4.1-mini with full description showed 0.829 similarity, while the output of GPT 4.1 mini with analogy-removed description only scored 0.814. This shows that context sample from GIFARC played a significant role in changing the task-solving approach of LLM to align with human-level analogic approach. Since GIFARC is a superset of the tested samples, it contains a much various types of analogies. Thus, this experiment shows the potential of GIFARC as the 'analogic approach' guiding dataset.

### 4.3 Performance Gains through GIFARC Fine-tuning

Through Section 4.1 and Section 4.2, we confirmed that GIFARC provides analogical reasoning guidance to models. In this section, we verify whether this analogical approach actually leads to improved performance in solving complex reasoning tasks in ARC-AGI. To quantitatively assess whether GIFARC improves reasoning capabilities beyond analogical understanding, we conducted supervised fine-tuning experiments on a language model. We then compared the performance of the GIFARC-trained model against the baseline model on the ARC-AGI.

We conducted comprehensive evaluations using both ARC-AGI-1 and ARC-AGI-2. The ARC-AGI-1 evaluation set contains 400 tasks with a total of 419 test pairs, while the ARC-AGI-2 set contains 120 tasks with 167 test pairs. To quantitatively verify whether GIFARC improves reasoning performance, we compared the performance of `Mistral-NeMo-Minitron-8B-Base` with the model fine-tuned on GIFARC on both evaluation sets. The fine-tuning was conducted using SFT with LoRA on 1,000 GIFARC tasks for 4 epochs.

We evaluated model performance using two metrics. First, we measured accuracy where a prediction is considered correct only when all elements in the predicted 2D grid exactly match the ground-truth grid; a single incorrect element results in a failed prediction. Second, we assessed model performance using Negative Log Likelihood (NLL), which measures how likely each model is to generate the correct grid. We computed NLL for all test pairs in each evaluation set. NLL is defined as $\text{NLL} = -\sum_{i=1}^{n} \log P(t_i | t_1, \ldots, t_{i-1}, \text{prompt})$ where $t_1, \ldots, t_n$ are tokens of the ground-truth response. NLL values indicate a higher likelihood of the model generating the correct answer, representing better performance.

Table 1 shows the quantitative evaluation results. The GIFARC-trained model demonstrates improvement across both metrics. In terms of accuracy, although both models failed to solve any tasks in ARC-AGI-2, reflecting its extreme difficulty, the ARC-AGI-1 results show more promise.

The dramatic decreases in NLL indicate that the model assigns significantly higher probability to correct answers after fine-tuning, suggesting improved understanding of the task structure. Although the absolute accuracy remains low due to ARC-AGI's stringent evaluation criterion, where even a single incorrect grid element results in task failure, both the accuracy gains on ARC-AGI-1 and the substantial NLL improvements across both evaluation sets demonstrate that the model is learning meaningful patterns and moving closer to correct solutions.

Table 1: Base and SFT model performance on ARC-AGI evaluation sets.

| Benchmark | Metric | Base Model | SFT Model |
|---|---|---|---|
| ARC-AGI-1 | NLL | 134 | 117 |
| | Accuracy | 0.2% (1/419) | 2.9% (12/419) |
| ARC-AGI-2 | NLL | 3724 | 204 |
| | Accuracy | 0% (0/167) | 0% (0/167) |

Quantitative analysis of Table 1 revealed that for ARC-AGI-1 evaluation, the model fine-tuned with GIFARC successfully solved additional problem types such as grid partitioning, object abstraction, and occlusion handling. For ARC-AGI-2 evaluation, problem types including object arrangement, object composition, and symbolic pattern application showed substantial NLL reductions. However, Figure 5 reveals that the current utilization of GIFARC is incomplete. Figure 5 shows that when examining the problem-solving process of the SFT model, there are cases where the model fails to detect appropriate analogies for tasks, and even when appropriate analogies are detected, they are not properly applied. These findings confirm that GIFARC is a useful learning resource, but suggest that more sophisticated learning methodologies are needed to improve analogy detection and application in order to maximize its effectiveness.

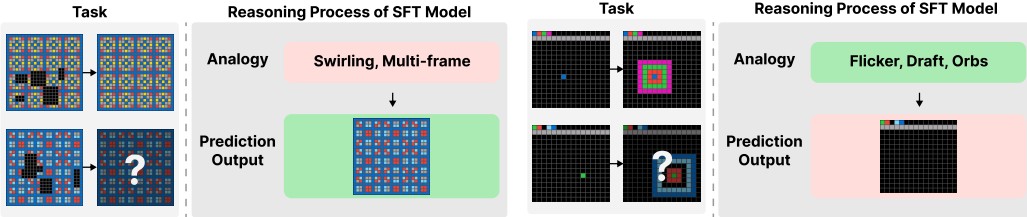

(a) Analogy detection failure          (b) Analogy application failure

Figure 5: Two failure modes in the SFT model's analogical reasoning. Figure 5a shows a case where the model produces the correct answer despite an incorrect analogy. The task requires identifying a repeating pattern and filling in black regions accordingly. However, the model incorrectly interprets this as a multi-frame sequence with a swirling effect. Figure 5b demonstrates a case where the model identifies an appropriate analogy but fails to apply it correctly. The model correctly reasons that the task involves creating stable orbs through flicker and draft effects, but cannot properly apply this analogy to generate the test output grid.

## 5 CONCLUSION

We introduced GIFARC, a large-scale synthetic ARC-style dataset grounded in human-intuitive analogies extracted from GIF images. GIFARC framework narrows the human-AI reasoning gap by embedding explicit analogy labels and executable solutions that help LLMs to map abstract grid transformations with everyday experiences. This approach has potential to improve AI-assisted education, scientific discovery, and decision-making support systems. Empirical results on the original ARC benchmark confirm that both fine-tuning on GIFARC and leveraging analogy cues in the reasoning step boost solver accuracy. However, our data generation pipeline is inherently dependent on a single GIF for analogy extraction, which places a limitation on the diversity and scope of analogies. In future work, we will scale GIFARC beyond motion-rich GIFs to videos and 3D simulations, extend evaluation to additional reasoning suites, and explore the fusion of multiple visual analogies to generate much more complex ARC tasks.

## REPRODUCIBILITY STATEMENT

Our work introduces a new dataset for leveraging human-intuitive analogies to elevate AI reasoning. To ensure reproducibility and promote further research, we have made the dataset publicly available at `https://huggingface.co/datasets/DumDev/gif_arc`. Pipeline codes used for generation and its instructions are available at `https://bit.ly/GIF-ARC`.

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

# APPENDIX

## A DATASET ANALYSIS

**Descriptive Statistics.** GIFARC currently contains 10,000 tasks, each accompanied by 103,357 train-test input/output pairs and an executable solution. The average input grid is 420±262 cells, while target grids are at 1,125±2,895. On average, a task uses 4.4±1.9 distinct colors, indicating that most tasks require multi-object reasoning or contain objects with some complexity, rather than binary foreground/background segmentation.

**Distribution of Task Types.** To analyze the semantic concepts and human-intuitive analogies represented in the dataset, we visualized the distribution of task types. Each keyword was mapped by GPT-4o into one of 20 coarse types, and Figure 6 presents the resulting histogram. Although the raw GIFs were initially sampled uniformly across 20 categories (e.g., nature, geometry, animation, ...), the distribution of task types in the final dataset diverges from this initial sampling. The generated problems cover a wide range of task types, with 'Rotational Symmetry & Perspective Spin' being the most dominant, and many tasks exhibiting a mixture of multiple types.

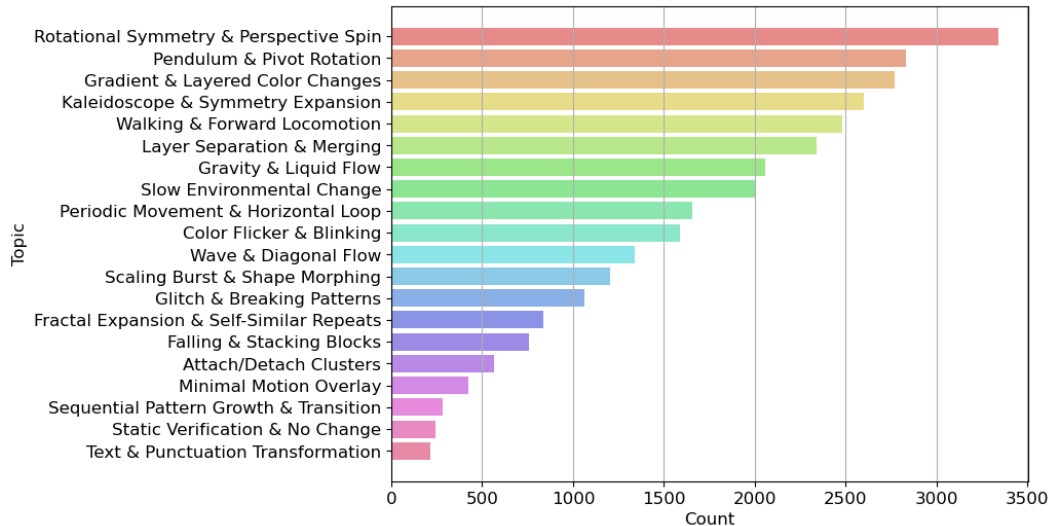

Figure 6: Histogram of task types occurrence in GIFARC. We classified the tasks in the ARC-AGI-2 training set using the 20 category tags defined in the GIFARC dataset.

**Comparing Taxonomy Distributions.** In the ARC-AGI-2 training dataset, Layer Separation & Merging accounted for the largest share (11.3%), followed by Gravity & Liquid Flow (9.6%) and Color Flicker & Blinking (9.3%), whereas in GIFARC, Rotational Symmetry is the dominant category. The average number of tasks per category was 36 (SD = 26). Although the distributions of GIFARC and ARC-AGI-2 differ, GIFARC achieves 90.7% category coverage. Moreover, because GIFARC translates diverse visual elements derived from GIFs into object concepts on the ARC grid and composes patterns from them, it can supplement task types that are absent from the ARC-AGI-2 training set.

**Task Complexity.** We assessed task complexity by measuring the complexity of the code used to generate GIFARC's grids. To this end, we report metrics such as the average number of lines of code (Lines of Code), the number of branching points in the code (Cyclomatic Complexity), the maximum depth of nested blocks in the code (Nesting Depth), and the number of unique operators used in the code (Unique Ops Count). These metrics make it easy to gauge how complex the processes required by GIFARC tasks are.

Code complexity metrics measured for GIFARC tasks are as follows: The average number of lines of code is 38.43±10.55, while the cyclomatic complexity averages 6.59±3.62. The maximum nesting

Table 2: Code Complexity of GIFARC Tasks compared to BARC.

| | GIFARC | | BARC | |
|---|---|---|---|---|
| | Mean | Std. | Mean | Std. |
| Lines of Code | 38.43 | 10.55 | 30.20 | 8.13 |
| Cyclomatic Complexity | 6.59 | 3.62 | 4.00 | 2.07 |
| Nesting Depth | 4.37 | 1.55 | 3.66 | 1.53 |
| Unique Ops Count | 7.03 | 0.28 | 7.42 | 1.07 |

depth in the code is $4.37\pm1.55$, and the number of unique operators averages $7.03\pm0.28$. These metrics quantitatively demonstrate that GIFARC contains tasks with diverse levels of complexity.

GIFARC has a longer average Lines of Code and a higher Cyclomatic Complexity score than BARC. The standard deviations of both metrics are also greater compared to BARC. This indicates that GIFARC encompasses problems with a relatively diverse range of code lengths and control flows, from simple branching to complex control structures. In contrast, Nesting Depth and Unique Ops Count does not show significant difference between the two datasets. Therefore, GIFARC can be considered a dataset that, compared to BARC, is similar to existing tasks in terms of nesting structures and variety of operations, while extending difficulty in code length and branching complexity - making it more complex and diverse in problem distribution than the original BARC.

**Generation Fidelity.** We measure the fidelity of our generation pipeline by evaluating the pass ratio at each stage. A task is considered a failure if it fails to compile or does not produce outputs in the expected format. Table 3 reports the success rates, showing that the overall pass ratio is consistently high across stages. The 19-20% failure rates mentioned are stage-specific figures, and since we only need to retry the failed stage rather than the entire pipeline, the actual additional API costs are very limited. Our analysis of the GIFARC data generation pipeline revealed two main failure causes: (1) API content policy-related errors and (2) non-conforming VLM responses, both of which can be addressed through simple retry logic and enhanced response validation.

Table 3: Success rate of generation.

| | Rate (%) |
|---|---|
| Visual abstraction stage | 94.12 |
| Task sketch stage | 80.54 |
| Executable ARC task stage | 81.82 |

**Qualitative Quality Evaluation** To evaluate whether the synthetically generated GIFARC tasks meaningfully incorporate the visual transformations extracted from the original GIFs, we conducted three human evaluation studies on 1,000 randomly sampled tasks. In our generation pipeline, step 2 produces task-defining concepts and descriptions that are explicitly designed to capture the core visual transformations present in the GIFs and to guide the synthesis of the final tasks. Human evaluators were then asked to assess whether the resulting synthetic ARC-style tasks faithfully reflected these step-2 concepts and descriptions. Our evaluation shows that 85.7% of the sampled tasks exhibit a clear alignment between the final task and its underlying concepts and descriptions. This suggests that, for the majority of GIFARC tasks, the dynamic visual elements captured at the concept-extraction stage are successfully integrated into the final task structure.

## B  EXAMPLES OF PROMPTS

### B.1  FULL PROMPT EXAMPLE FOR EXTRACTING VISUAL ABSTRACTIONS

The extraction of visual abstractions constitutes the first step in the data genera-
tion process.   This process involves extracting comprehensive information from GIFs,
including `visual_elements`, `objects`, `static_patterns`, `dynamic_patterns`,
`core_principles`, and `interactions`. The specific content requirements for these six types
of information and the response format for the VLM are detailed in both the user prompt and system
prompt below.  The following content presents the complete specification of the aforementioned
prompts.

---

**User Prompt: Extracting Visual Abstractions from GIFs**

Please analyze the given GIF by extracting core reasoning elements. For each section below, provide
your response as a list of clearly separated items (not long-form paragraphs).

**[visual_elements]**

- Identify the key visual elements, objects, and actors in the GIF.
- List the major visual components in the scene (e.g., objects, colors, spatial arrangement,
  shapes).

**[objects]**

- List all visual objects appearing in the GIF using this format:

```
{
  "name":  "object name",
  "type":  "explicit" or "implicit"
}
```

- explicit = physical objects with clear boundaries
- implicit = patterns or structures formed by multiple elements
- Identify 2-8 key objects.

**[interactions]**

- List object interactions using this format:

```
{
  "objects_involved":  ["object1", "object2"],
  "interaction_type":  "clear/ambiguous/constraint",
  "interaction_parameters":  ["parameter1", "parameter2"]
}
```

- `interaction_type` definitions:
  - **clear**: distinct physical interactions (collisions, contact, etc.)
  - **ambiguous**: indirect or unclear interactions
  - **constraint**: interactions that establish limitations or boundaries
- Record 2-6 key interactions, including only those directly observed in the GIF.

**[static_patterns]**

- Identify the elements or relationships that remain consistent throughout the GIF.
- Describe repeating spatial arrangements, consistent backgrounds, or fixed design features.
- List all patterns or objects that remain constant throughout the GIF.

**[dynamic_patterns]**

- Describe how the elements interact and change over time.
- Consider whether the changes occur gradually (step-by-step), abruptly, or through multi-stage
  transformations (e.g., directional shifts, scaling, rotation).
- List the distinct changes or interactions that occur over time.

**[core_principles]**

---

- Identify all general reasoning principles or mechanisms that explain **how and why** the dynamic elements change over time.
- Each principle should be **generalizable beyond the current GIF** and help abstract the reasoning structure behind the transformation.
- List one or more such principles that account for the observed dynamics.
- After listing them, **summarize the single most fundamental principle in one concise sentence.**
- Examples include: physical forces (e.g., gravity causes downward movement), goal-oriented behaviors (e.g., movement toward a target), causal chains (e.g., one event triggers another), symmetry-based transformations (e.g., reflection or alignment), and repetitive or cyclic patterns.

---

**System Prompt: Extracting Visual Abstractions from GIFs**

You are an analysis assistant. Your role is to extract structured and reproducible information from a visual GIF using the five categories below.

You **MUST** focus on observable phenomena, transformation structures, and etc.

Do **NOT** invent unobservable events or make abstract generalizations. Describe only what can actually be seen in the GIF.

For each category below, provide a plain list of items (one per line or bullet point). Avoid paragraph-style narration. Focus only on concrete, observable phenomena and transformation patterns.

Return your response strictly in the following JSON format:

```
{
  "visual_elements": [
    "<list of observable objects, colors, spatial arrangements, or
notable visual traits>"
  ],
  "objects": [ {
      "name": "<object name>",
      "type": "explicit/implicit"
  } ],
  "interactions": [ {
      "objects_involved": ["<object1>", "<object2>"],
      "interaction_type": "clear/ambiguous/constraint",
      "interaction_parameters": ["<parameter1>", "<parameter2>"]
  } ],
  "static_patterns": [
    "<list of all objects or properties that remain unchanged
throughout the GIF>"
  ],
  "dynamic_patterns": [
    "<list of all distinct transformations or movements that occur
over time>"
  ],
  "core_principles": [
    "<list of general reasoning principles behind the transformation
(e.g., gravity causes vertical motion)>"
  ]
}
```

## B.2 FULL PROMPT EXAMPLE FOR TASK SKETCH

The task sketch represents the second step in the data generation process. The objective of this step is to design ARC-style tasks by dividing them into `concepts` and `description` components, utilizing the six types of information extracted in Step 1. The user prompt provides the detailed specifications for this process.

---

**User Prompt: Task Sketch**

You've generated these on previous requests:
```
{examples}
```

Based on your previous GIF analyses, I'd like you to create a new concept that incorporates the key elements from this analysis:
```
objects:  {objects}
static_patterns:  {static_patterns}
dynamic_patterns:  {dynamic_patterns}
interactions:  {interactions}
core_principles:  {core_principles}
```

Brainstorm one more.

If the above informations involve gradual changes over time or a process of reaching a clear goal state, the input and output grids should not merely depict a short-term or single-frame transition. Instead, they should be designed to capture the entire transformation process from the initial state to the final state.

When temporal progression or cumulative change is central to the analogy, construct the task code so that this flow and its outcome are clearly reflected through the difference between input and output.

In this case, please create concepts and descriptions that effectively incorporate the above visual elements, static patterns, dynamic patterns, and core principles, following the format below, while referencing previous examples:
```
# concepts:
# <concepts in your new generation>

# description:
# <description of your new generation>
```

---

## B.3 Full Prompt Example for Generating Executable ARC Tasks

We generated ARC-style tasks using the analogies extracted from Step 1 and Step 2. Step 3 utilizes original BARC seeds that were used to generate input and output grid pairs. The following prompt contains the detailed explanation for Step 3.

---

**User Prompt: Puzzle Implementation from Description**

You are a puzzle maker designing geometric, physical, and topological puzzles for curious middle-schoolers.

Each puzzle consists of uncovering a deterministic rule, pattern, procedure, algorithm, or transformation law that maps inputs to outputs. Both the inputs and outputs are 2D grids of colored pixels. There are 10 colors, but the order of the colors is never relevant to the puzzle.

The middle schoolers are trying to discover this deterministic transformation, which can be implemented as a Python function called `main`. Designing a puzzle involves also creating example inputs, which can be implemented as a Python function called `generate_input`. Unlike `main`, the `generate_input` function should be stochastic, so that every time you run it, you get another good example of what the transformation can be applied to.

Here is a overview of the puzzle you are designing:

`{description}`

Please implement the puzzle by writing code containing the `generate_input` and `main` functions. Use the following standard library (`common.py`):

`{common_lib}`

Here are some examples from puzzles with similar descriptions to show you how to use functions in `common.py`:

`{examples}`

Your task is to implement the puzzle, following these steps:

1. Inspect the example puzzle implementations, making note of the functions used and the physical/geometric/topological/logical details
2. Inspect the new puzzle's description
3. Brainstorm a possible implementation for the new puzzle
4. Generate a code block formatted like the earlier examples with a comment starting `# concepts:` listing the concepts and `# description:` describing the inputs and transformation from the given description.

When implementing code, please avoid using float type variables, numbers with decimal points, and the math library. These elements make it difficult to intuitively identify patterns between input and output grids. Instead, use only integer operations and basic arithmetic operators to clearly reveal the essence of the pattern. Please strictly follow this constraint when implementing your code.

Also, When implementing code, please use the minimum number of lines possible. As code gets longer, its complexity increases, and if it becomes too detailed and complicated, people will find it difficult to intuitively understand the puzzle's rules just by looking at the input and output grids. Situations where one needs to analyze the code to understand the rules should be avoided. Please write concise and efficient code that clearly reveals the core pattern.

Be sure to make the transformation `main` deterministic. Follow the description closely.

---

**System Prompt: Generating Executable ARC Task (Version 1)**

You need to help me write code containing the `generate_input` and `main` functions according to the given puzzle design. You must use the standard library (`common.py`). Create an appropriate puzzle following the given puzzle design concepts and description.

When writing code, please use variable names that are meaningfully related to the core concepts of the problem. For example, if the problem involves snow falling phenomena, use variable names like `snowflake`, `precipitation`, `accumulation`, `gravity`, `obstacle`, etc. Specifically, when implementing the `generate_input` function and `main` function, make sure each variable name is directly associated with the concepts in the problem. For instance, use `gravity_strength` for a variable representing the intensity of gravity, and `obstacle_positions` for storing the locations of obstacles — choose names that clearly reveal the role and meaning of each variable.

Additionally, in the `generate_input` function, please restrict the grid size to be between `1x1` and `30x30`. Do not create grids larger than `30x30`. Implement the `generate_input` function that creates inputs appropriate for the problem and the `main` function that utilizes them while following these constraints.

When implementing code, please avoid using float type variables, numbers with decimal points, and the math library. These elements make it difficult to intuitively identify patterns between input and output grids. Instead, use only integer operations and basic arithmetic operators to clearly reveal the essence of the pattern. Please strictly follow this constraint when implementing your code.

Also, when implementing code, please use the minimum number of lines possible. As code gets longer, its complexity increases, and if it becomes too detailed and complicated, people will find it difficult to intuitively understand the puzzle rules just by looking at the input and output grids. Situations where one needs to analyze the code to understand the rules should be avoided. Please write concise and efficient code that clearly reveals the core pattern.

When doing this, please output your solution following the JSON format specified below.

```
{
  "library": "<Write only the libraries used in the code.
Ex.  from common import* \n import numpy as np \n ....>",
  "main_code": "<Write the main code part.>",
  "generate_input_code": "<Write the generate input code
part.>",
  "total_code": "<Write total code including libraries,
main, generate_input and given concepts and description>"
}
```

## B.4 FULL PROMPT EXAMPLE FOR GENERATING ANALOGY GIVEN TASKS

In the Section 4.2, we used the following prompt to generate analogies for the given task.

---

**User Prompt: Generating Analogy**

**TOP-LEVEL INSTRUCTIONS – READ FIRST**

- The section between **Few-Shot Examples** and `<task>` is an example block that demonstrates how to solve the task using analogies.
- The lines that follow `<task>` present the task you must solve, consisting of Training Examples and Test Examples.

**Few-Shot Examples**

```
<additional_information>
This information provides helpful context to solve the given ARC
task.

<description w/ analogy or description w/ analogy and solution -
1>
...
<description w/ analogy or description w/ analogy and solution - 15>

<task>

</task>

<additional_information>
This information provides helpful context to solve the given ARC task.

{description w/ analogy or description w/ analogy and solution - 1}
...
{description w/ analogy or description w/ analogy and solution - 15}

</additional_information>

<task>
...
</task>
```

Please solve task thoroughly and structure your response with the following sections:

**# THINKING ANALOGY**

```
[Let's think the analogical pattern about task step by step]
Let's think step by step about the analogical pattern in the task, and
describe what kind of analogical, metaphorical, or figurative pattern
it is
```

**# THINKING PROCESS**

```
[Let's think the task step by step by leveraging thinking analogy and
task analogy]
Let's think the task leveraging the analogical, metaphorical, or figu-
rative patterns observed in the 15 examples to uncover and apply the
underlying analogy
```

**# SOLUTION CODE**

```
[Write the final solution code for the task based on the analogical
approach identified in the thinking process]
```

Let's solve the problem step by step.

---

# C IMPLEMENTATION DETAILS

## C.1 MODEL HYPERPARAMETERS

In this pipeline, we set the temperature to 0 in order to build deterministic datasets. Note that reasoning models like OpenAI o-series do not support temperature configuration, so this parameter was not set for those models. For more detailed hyperparameters, refer to Table 4.

Table 4: OpenAI API Hyperparameters

| Process Name | Role | Model Name | Max Tokens | Top p | Temperature |
|---|---|---|---|---|---|
| Method Step 1 | Data Generator | o4-mini | 2,048 | 1.0 | - |
| Method Step 2 | Data Generator | o4-mini | 2,048 | 1.0 | - |
| Method Step 3 | Data Generator | o4-mini | 2,048 | 1.0 | - |
| Application | LLM Evaluator | o3-mini | 40,000 | 1.0 | - |
| Application | LLM Solver | GPT 4.1-mini | 32,768 | 1.0 | 0.0 |

## C.2 DATA GENERATION

GIFARC builds on the data generation process established by the prior work BARC (Li et al., 2024). The original BARC framework generated tasks using manually crafted functions called seeds. A key difference from BARC is that GIFARC employs o4-mini to extract six key pieces of information from GIFs in Step 2, which are then used to design tasks in Step 3. Generation is based on the `Description`. Based on the designed tasks, we use o4-mini to generate code in Step 3 that creates inputs and the `main` function that produces outputs. We create input and output grids and apply a filtering process using this generated code.

The filtering criteria consist of ten conditions adapted from BARC, as shown in Table 5. Input-output grid pairs that meet any of these conditions are filtered out. In GIFARC, creating tasks that incorporate analogies often results in complex code. Therefore, we set a maximum time limit of 300 seconds per task to provide sufficient processing time. Cases requiring more time are considered to be caught in infinite loops and are filtered out as timeout failures. This filtering process aims to enhance the quality of GIFARC-generated tasks.

Table 5: Filtering Conditions

| Filtering Condition | Meaning |
|---|---|
| Non-Deterministic | The output grid differ when the same transformation is applied multiple times to the same input. |
| Non Color Invariant | 1. Cases where the transformation function itself fails during the color permutation process. 2. Either the permuted grid or the input grid is not well-formed. 3. The results differ even when only the colors are changed. |
| Identity | The input and output are completely identical. |
| Non-Well Formed Output | The transformation result is not well-formed (i.e., not a 2D list with equal row lengths and integer values between 0-9). |
| Black Output | The output consists entirely of 0s (black pixels). |
| Timeout | Overall time limit exceeded. |
| Non-Well Formed Input | The input is not well-formed (i.e., not a 2D list with equal row lengths and integer values between 0-9). |
| Duplicate Input | The generated input is duplicated with existing ones. |

## C.3 APPLICATION

We conducted experiments on Section 4.1 which is "Verification Changes in Reasoning Steps" and Section 4.2 which is "Alignment between Context-Generated and Ground-Truth Analogies". For these experiments, we developed various prompt context conditions: full description, description without analogy, and description without analogy and without solution.

The full description was generated using o3-mini to enhance the analogies in both the $\alpha$ and $\phi$ of the GIFARC dataset. To prevent the inclusion of irrelevant analogies, we performed manual refinement of the generated content. For the description without analogy, we utilized o3-mini to remove analogy-related information from the full description. Subsequently, we converted any remaining analogical elements in both $\alpha$ and $\phi$ to grid-level representations. The description without analogy and without solution used the same analogical framework as the description without analogy. Across all kinds of descriptions, the grid information remained consistent.

In our experiments examining Section 4.1 and Section 4.2, we employed both the full description and the analogy-removed description. The full description was identical to that used in the Section 4.1 application. The analogy-removed description was a refined version of the description without analogy used in the Section 4.1 experiment, where we completely eliminated any remaining analogical information by either removing it entirely or converting it to grid-level representations, thus ensuring the complete absence of analogical content.

## D  GUIDELINES FOR HUMAN EVALUATORS ASSESSING THE GIVEN TASKS

In the section 4.2, we conducted human evaluation on 12 tasks. Three experts familiar with ARC were asked to describe the analogies for 12 GIFARC tasks. To obtain responses of sufficient quality for our experiment, we provided minimal guidelines. The following is a brief outline of the guidelines provided to the human evaluators.

---

**Guidelines for Human Evaluators**

Please write the pattern analogy in English sentences for the provided 12 tasks (input-output pairs), referring to the content below:

1. What the provided input-output pairs are visualizing
2. What kind of analogy the pattern changing from input to output can be related to or expressed as
3. What the colors or object shapes shown in the input/output symbolize or represent
4. Please write the expected rules if you don't know the task's analogy.
Ex-1) - It appears to be spreading from the center.
Ex-2) - The movement seems to change randomly and the colors are changing.

[Working Example]
{Example task}

Analogy: In this task the grid represents a scene where small buds (colored gray: 5) transform into blooming flowers. The transformation simulates the process of blooming: each bud expands its presence by turning its neighboring cells (up, down, left, right) into petal cells (colored gray: 5) while the original bud cell becomes the highlight of the bloom (colored yellow: 4).

[GIFARC Task 1]
{Input-Output Pairs}

$\vdots$

[GIFARC Task 12]
{Input-Output Pairs}

---

Based on this guideline, participants described the analogies for the given GIFARC tasks. The information collected contained only content about the analogies, and no personal or sensitive information was gathered.

# E   EXAMPLE VISUALIZATIONS OF GIFARC

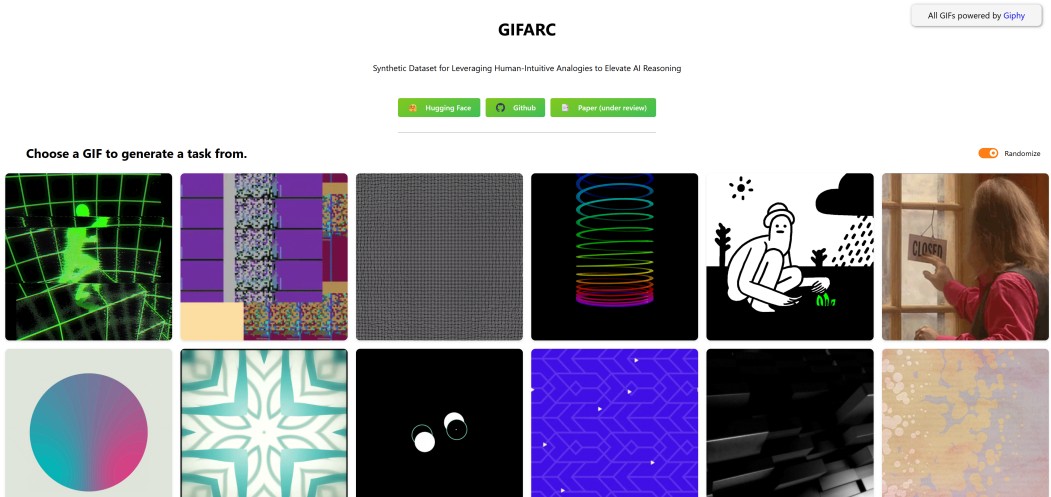

Figure 7: Landing Page of GIFARC Visualization website. It is publicly released at `https://gifarc.vercel.app/`

All 10,000 tasks generated with GIFARC are visualized in `https://gifarc.vercel.app/`. Following are example visualizations of GIFARC-generated ARC-style tasks of each task type.

## E.1   EXAMPLE TASK OF TYPE 1 - ROTATIONAL SYMMETRY & PERSPECTIVE SPIN

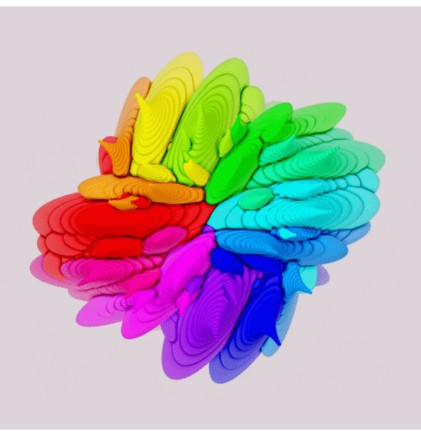

Figure 8: Snapshot of GIF used for generating task 962.

- Concepts : fractal expansions, symmetrical pulsations, radial transformations, iterative growth

- Description : In the input, you will see a sequence of grids (frames) on a black background. Each grid depicts: - A large circular region in the center, with multiple concentric rings and radial lines diverging outward. - A stacked-curve fractal anchored at the bottom-left corner. - A set of spire fractals anchored at the bottom-right corner. As you move from one frame to the next in the input, these elements undergo iterative transformations: - The radial lines repeatedly shift in thickness and visual intensity, while preserving rotational symmetry. - The concentric rings in the circular region pulsate by alternately expanding and contracting. - The stacked-curve fractal on the bottom-left changes its curve density, fractally adding or removing segments. - The right-side spire fractals expand and contract in repeated vertical segments. Your task is to replicate and apply these

transformations for the entire sequence, and produce the final frame of the animation as the output: 1) Ensure the anchoring of the fractals at the bottom edge remains the same. 2) Preserve the radial symmetry of the central circular region and its common center point. 3) For each pulsation step in the input, magnify or contract the rings, lines, and fractals accordingly. 4) Continue until the final pulsation step is reached. That final state is your output grid. The essential principle is that symmetry-based fractal transformations and repeated cyclical expansions/contractions produce the overall pulsating effect. By reflecting each iterative change step by step, you reconstruct the final pulsating pattern visible at the end of the sequence.

- Full web view is available at `https://gifarc.vercel.app/task/962`.

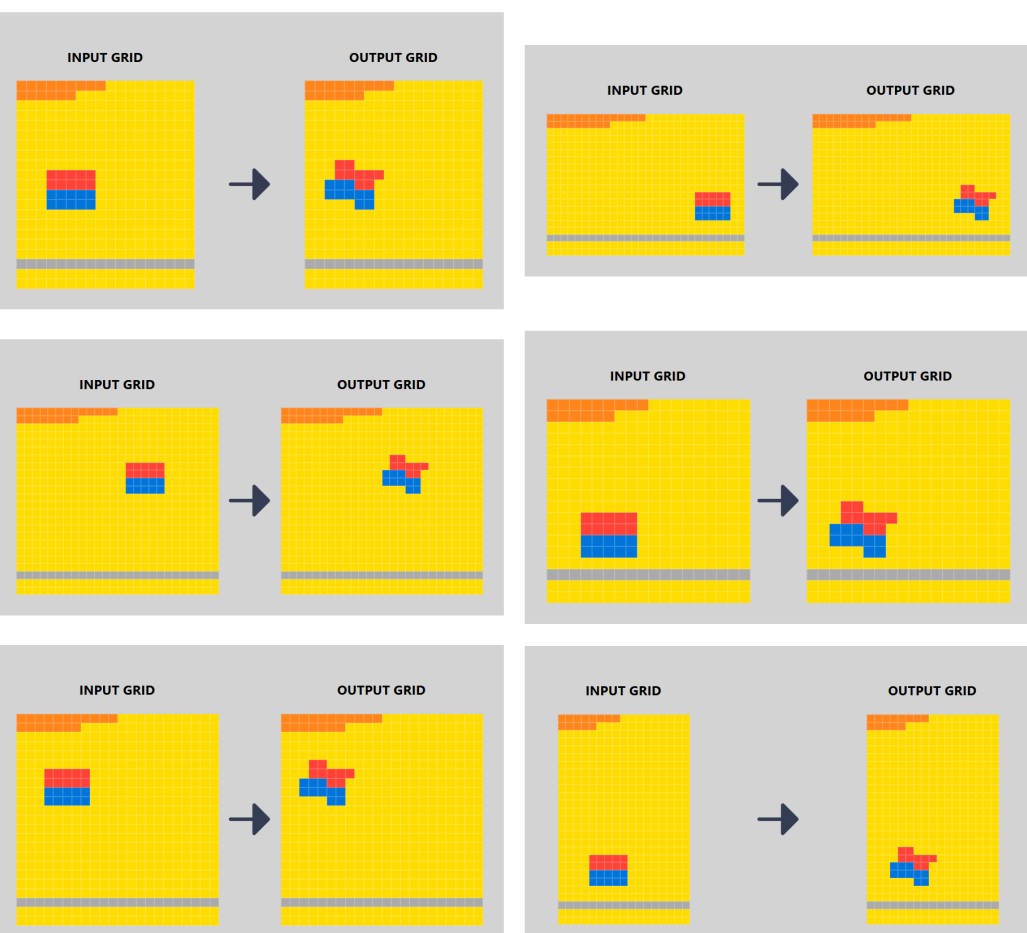

Figure 9: GIFARC-generated task 962.

Listing 1: Python code solution for task 962.

```python
from common import *
import numpy as np
import random

# concepts:
# rotational motion, center-of-mass shifting, multi-step transformation,
    static background

# description:
# The input grid shows a paved surface (Color.YELLOW) with static
    features: a fence (Color.GREY) drawn near the bottom and mountains
    (Color.ORANGE) at the top.
# A dynamic object-a wheelchair with its occupant-is represented by a
    4x5 sprite: the top two rows are the occupant (Color.RED) and the
    bottom two rows the wheelchair (Color.BLUE).
# The wheelchair rotates around an axle (computed as the average
    position of the blue pixels) so that the occupant's mass shifts
    further off support.
# The transformation in main erases the original dynamic pixels and
    re-blits them using an integer-approximated clockwise rotation
    (about 22 degrees) around the axle, while leaving the static
    background unchanged.

def main(input_grid):
    out=input_grid.copy()
    dyn=[]
    for i in range(len(input_grid)):
        for j in range(len(input_grid[0])):
            if input_grid[i,j] in [Color.BLUE,Color.RED]:
                dyn.append((i,j,input_grid[i,j]))
                out[i,j]=Color.YELLOW
    if not dyn: return out
    ax_i=sum(p[0] for p in dyn)//len(dyn)
    ax_j=sum(p[1] for p in dyn)//len(dyn)
    cos_val, sin_val = 92,38
    for i,j,col in dyn:
        di=i-ax_i; dj=j-ax_j
        ni=ax_i+((cos_val*di+sin_val*dj)//100)
        nj=ax_j+((-sin_val*di+cos_val*dj)//100)
        if 0<=ni<len(out) and 0<=nj<len(out[0]): out[ni,nj]=col
    return out

def generate_input():
    H,W=random.randint(15,30),random.randint(15,30)
    grid=np.full((H,W),Color.YELLOW)
    for j in range(W): grid[H-3,j]=Color.GREY
    for i in range(2):
        for j in range(W//(2+i)):
            grid[i,j]=Color.ORANGE
    sprite=np.full((4,5),Color.BLACK)
    sprite[:2,:]=Color.RED; sprite[2:,:]=Color.BLUE
    x,y=random_free_location_for_sprite(grid,sprite,background=
        Color.YELLOW)
    blit_sprite(grid,sprite,x,y)
    return grid
```

### E.2 EXAMPLE TASK OF TYPE 2 - KALEIDOSCOPE & SYMMETRY EXPANSION

- Concepts: kaleidoscopic transformation, mirrored folding, cyclical animation, vertical symmetry

- Description : In this puzzle, the input is a multi-frame sequence depicting a pale-yellow corridor-like structure with a central rectangular opening and mirrored arches aligned along a vertical axis of symmetry. Over several frames, the corridor's walls fold inward and outward in a smooth, kaleidoscopic motion, always returning to the initial configuration. The output is the full cycle of transformations repeated in a loop, preserving the corridor's pale-yellow hue and central frame while continuously reflecting the mirrored arch patterns around the vertical axis. This cyclical approach showcases the corridor morphing into itself, completing each transformation stage and then re-starting at the original symmetrical configuration.

- Full web view is available at `https://gifarc.vercel.app/task/7442`.

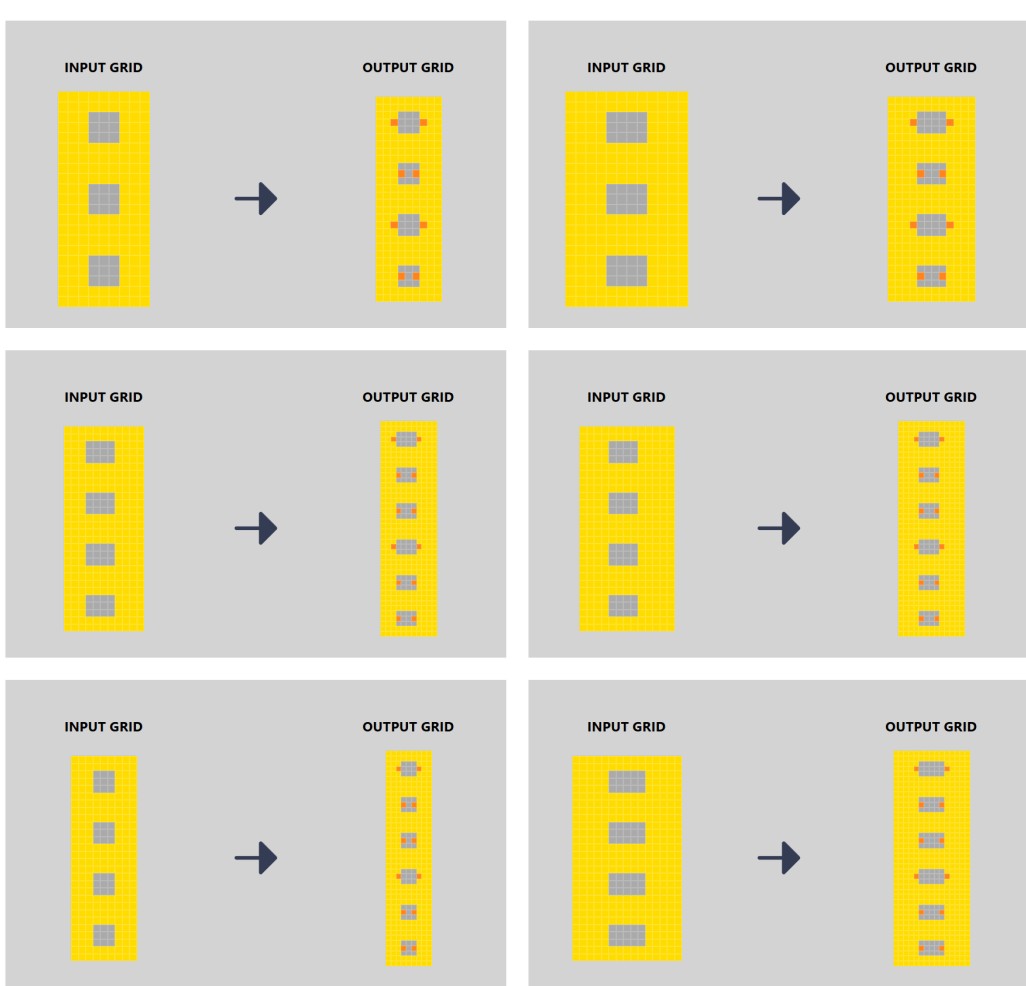

Figure 10: GIFARC-generated task 7442.

### E.3 EXAMPLE TASK OF TYPE 3 - PENDULUM & PIVOT ROTATION

- Concepts : cyclical rotation, pivot anchoring, animation frames

- Description : Inspired by previous examples that repeated or translated objects across multiple frames, in this puzzle the input is a single snapshot showing a blue curved shape on a black background. For the output, construct a multi-frame grid depicting the same shape completing a full 360-degree rotation around a central pivot. Each frame rotates the shape by a fixed angle (e.g., 15° increments), and the frames are arranged in a 3x3 or 4x4 grid until the shape returns to its original orientation. The black background and the blue color remain consistent across frames, and the shape never goes outside the canvas boundary. This cyclical rotation from start to finish, then repeating, is the puzzle's core concept.

- Full web view is available at https://gifarc.vercel.app/task/1037.

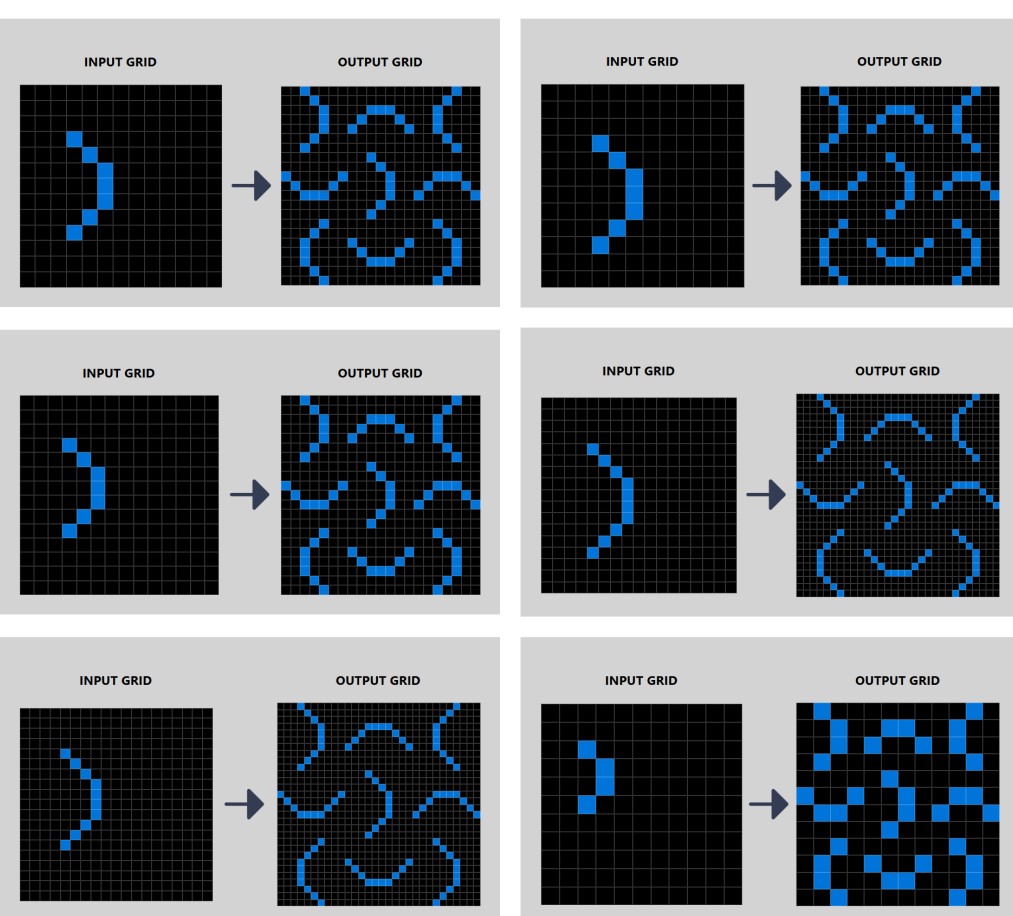

Figure 11: GIFARC-generated task 1037.

### E.4 EXAMPLE TASK OF TYPE 4 - WALKING & FORWARD LOCOMOTION

• Concepts : cyclical motion, repetition, background constraint

• Description : In the input you will see an animation (or multiple frames) of pink human-shaped figures in a light-blue background. White wavy lines span the background and shift subtly as the pink figures travel diagonally from the top-left to the bottom-right. When a pink figure leaves the bottom-right edge, it reappears at the top-left edge at the same vertical position, making a continuous loop. The translucent blue overlay patterns remain visually consistent on top of the background while the white lines gently oscillate as the figures pass. To produce the output, replicate this cyclical motion fully at least once, preserving continuity. The final frame must line up so that if we start again from there, the pink figures continue repeating the same path through the light-blue background in a seamless loop.

• Full web view is available at `https://gifarc.vercel.app/task/2767`.

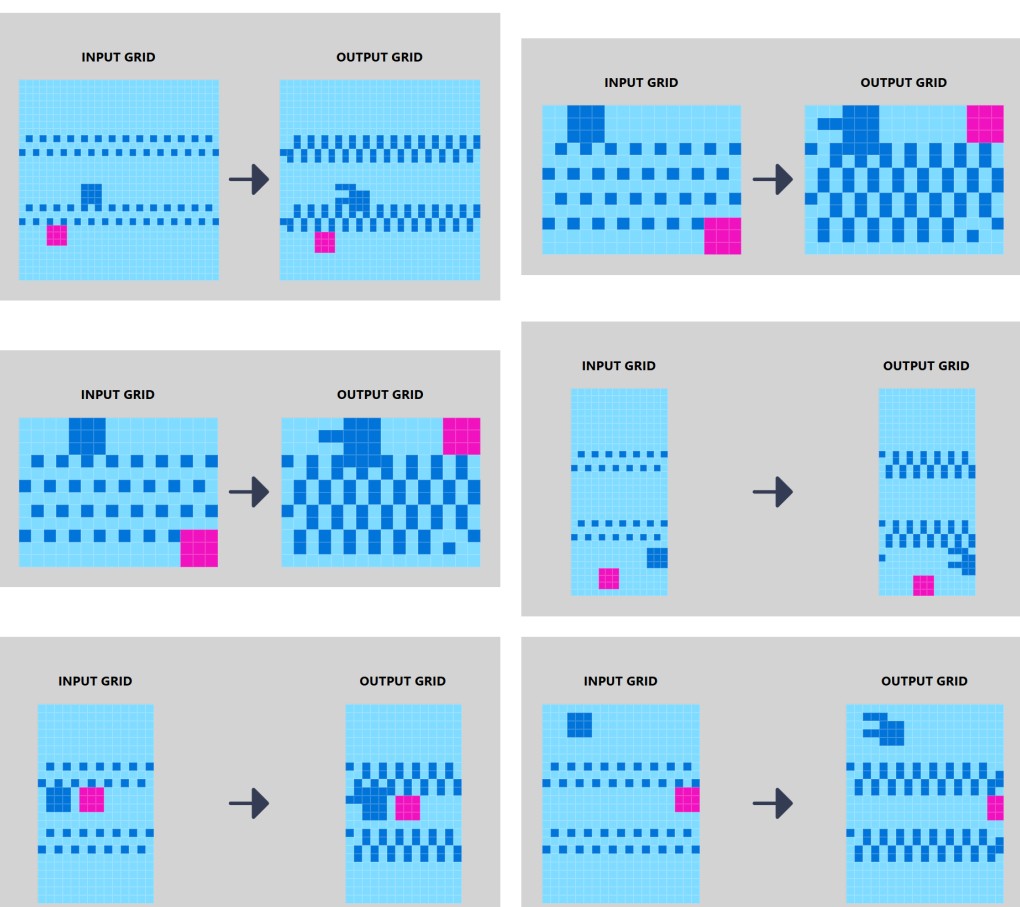

Figure 12: GIFARC-generated task 2767.

## E.5 EXAMPLE TASK OF TYPE 5 - FALLING & STACKING BLOCKS

- Concepts : handheld device, screen area, gravity, falling blocks, stacking, looping animation

- Description : In the input, you will see a pink background with a grey handheld gaming device in the center. The device has a bright green rectangle acting as its screen, with a d-pad on the left and action buttons on the right. Initially, you will see Tetris-like blocks of various shapes hovering above the upper edge of this green screen area. To create the output, you must simulate the blocks falling downward into the screen, stacking on top of one another or on previously fallen blocks. Keep each block fully within the green screen boundary; if a falling block collides with another block or the bottom of the screen, it must stop. Once the screen is completely filled to the top, the stack disappears, effectively resetting the screen to empty and repeating the cycle. In other words, show the progression from "empty screen" → "stacked blocks" → "reset," producing a looping effect of the falling Tetris-like blocks within the green screen.

- Full web view is available at `https://gifarc.vercel.app/task/556`.

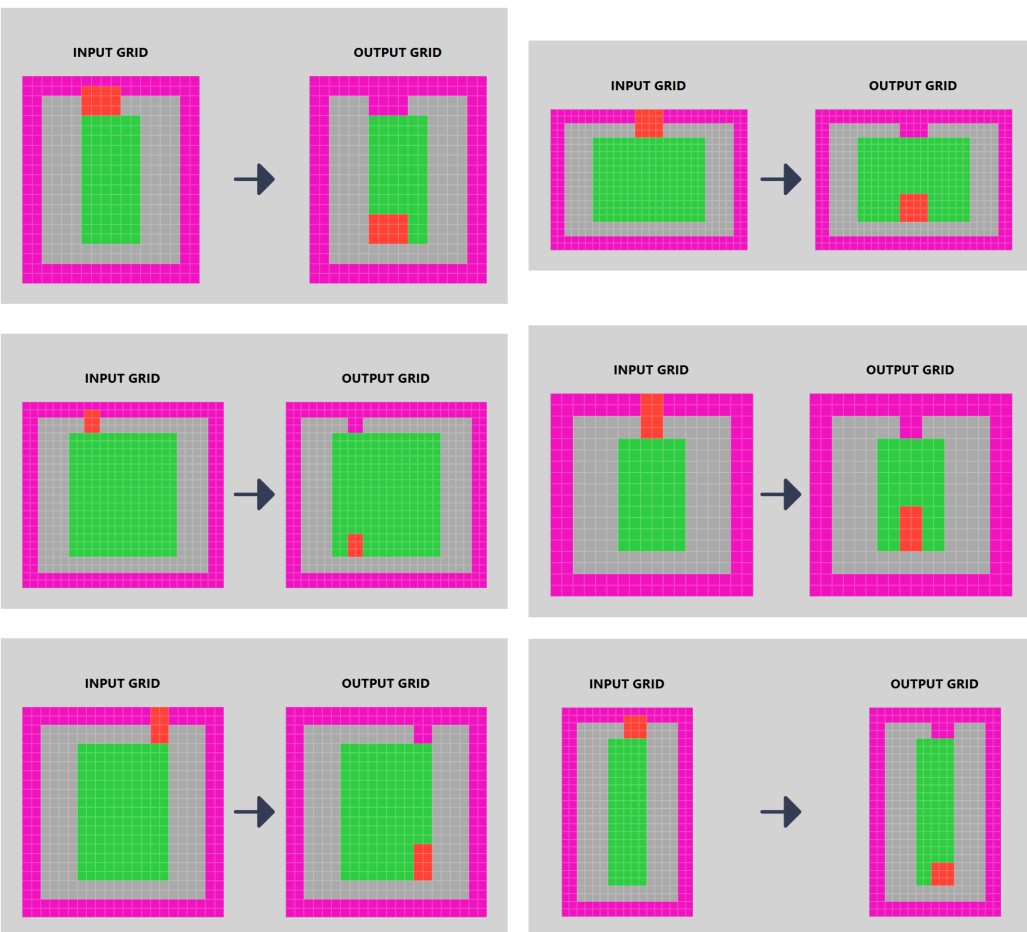

Figure 13: GIFARC-generated task 556.

### E.6 EXAMPLE TASK OF TYPE 6 - PERIODIC MOVEMENT & HORIZONTAL LOOP

- Concepts : cyclical motion, frame-based animation, horizontal repetition, background preservation

- Description : In the input you will see several frames depicting a single brown triceratops, side-profile, walking across a beige background. The frames show a gradual shift in the triceratops' position from left to right while its legs cycle through a walking gait. To create the output, stack or tile these frames into a continuous strip (or series) that repeats seamlessly. The background stays beige and does not change between frames, and the triceratops remains the same brown color and orientation. The final output shows a horizontally looping animation strip where the last frame transitions smoothly back into the first, preserving the cyclical walking motion and the uniform background.

- Full web view is available at https://gifarc.vercel.app/task/3690.

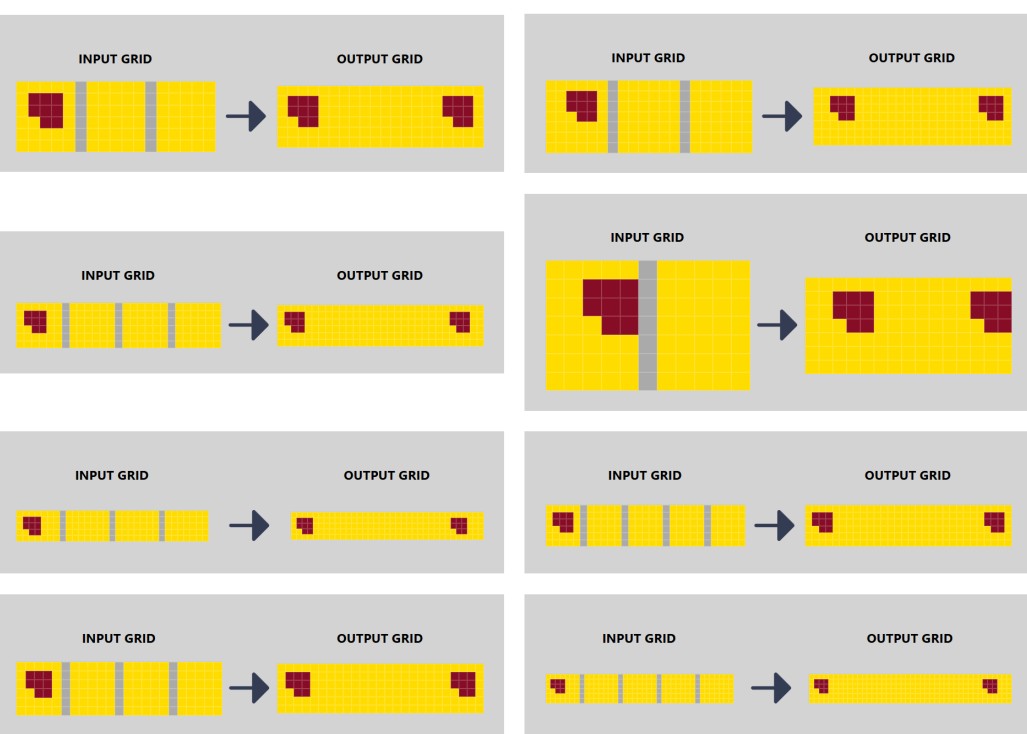

Figure 14: GIFARC-generated task 3690.

### E.7 EXAMPLE TASK OF TYPE 7 - COLOR FLICKER & BLINKING

- Concepts : cyclical color flicker, hinge constraints, multi-frame puzzle, looping animation

- Description : The input consists of multiple frames showing a hand-drawn scene: a large circular shape with concentric scribbles/colors in its center, topped by a smaller hinged circle, with various scribbles at the right (ladder-like), left (darker scribble), and near the hinge (green). Over time, the concentric scribbles in the center circle flicker through different color states while all objects remain in the same relative positions.

- Full web view is available at `https://gifarc.vercel.app/task/385`.

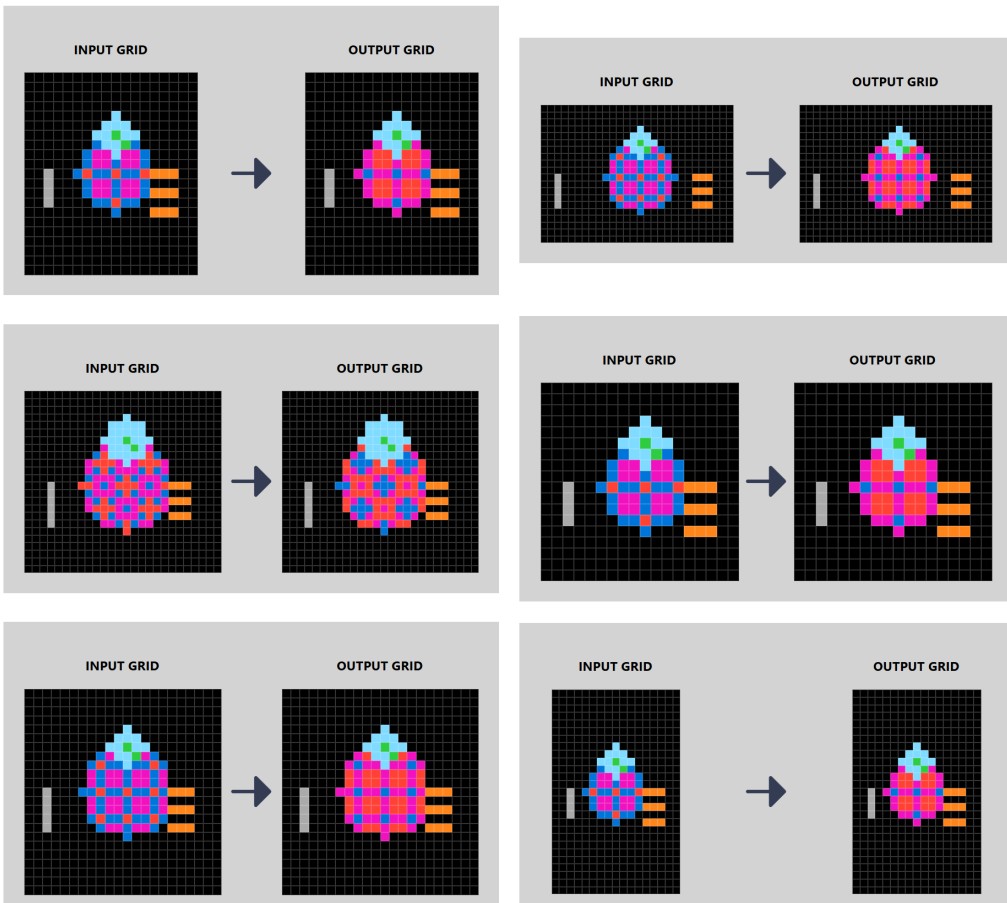

Figure 15: GIFARC-generated task 385.

### E.8 EXAMPLE TASK OF TYPE 8 - GRADIENT & LAYERED COLOR CHANGES

- Concepts : radial color cycling, symmetry, continuous transformations, layering
- Description : In the input, you will see a single circular shape on a purely black background. Inside the circle is a radial, symmetrical pattern composed of multiple colors. Each concentric ring of the circle shifts colors in a periodic cycle, from an inner ring (earliest phase) to an outer ring (later phase), repeatedly. The circle retains its size and position, and the background remains black.
- Full web view is available at `https://gifarc.vercel.app/task/117`.

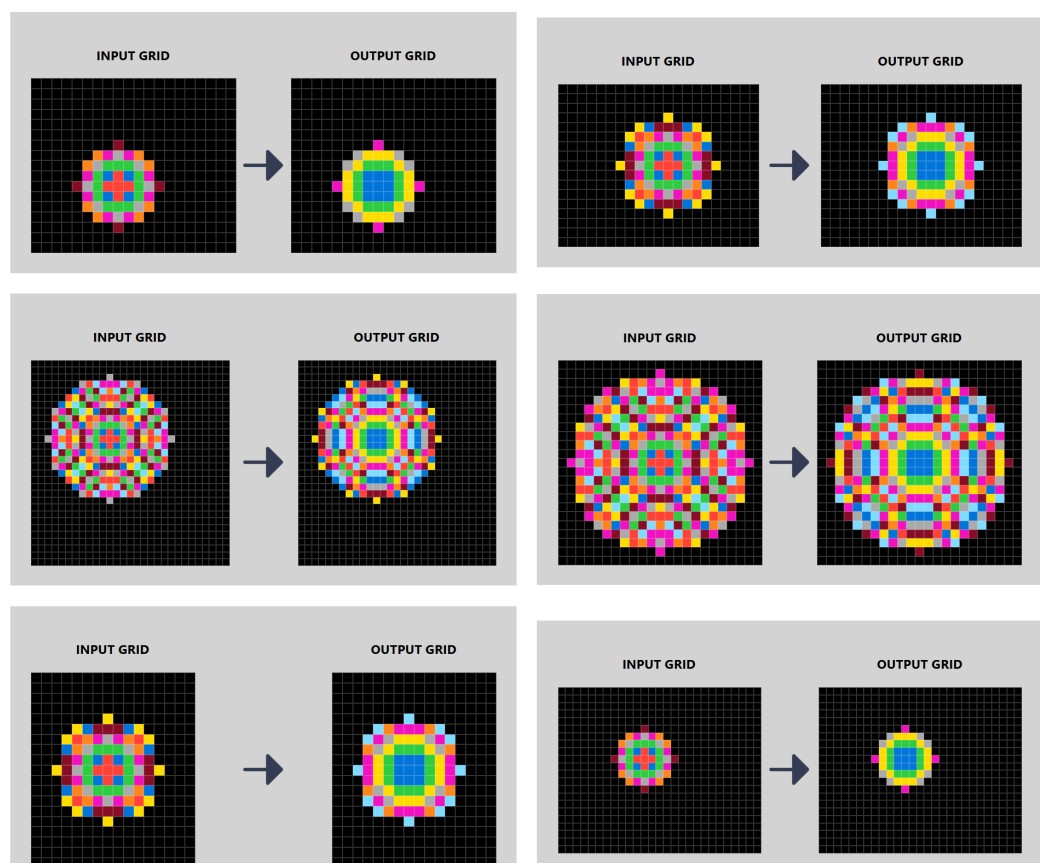

Figure 16: GIFARC-generated task 117.

## E.9 EXAMPLE TASK OF TYPE 9 - GLITCH & BREAKING PATTERNS

- Concepts : glitch patterns, color flickering, repetitive animation, partial stability
- Description : In the input you will see multiple horizontal bars stacked vertically over a black background. Each bar is composed of green, purple, and yellow pixels arranged in rectangular stripes. Some small clusters of yellow pixels flicker like digital noise, intensifying toward the center and right side of each bar. The bars remain in the same positions throughout the sequence.
- Full web view is available at `https://gifarc.vercel.app/task/410`.

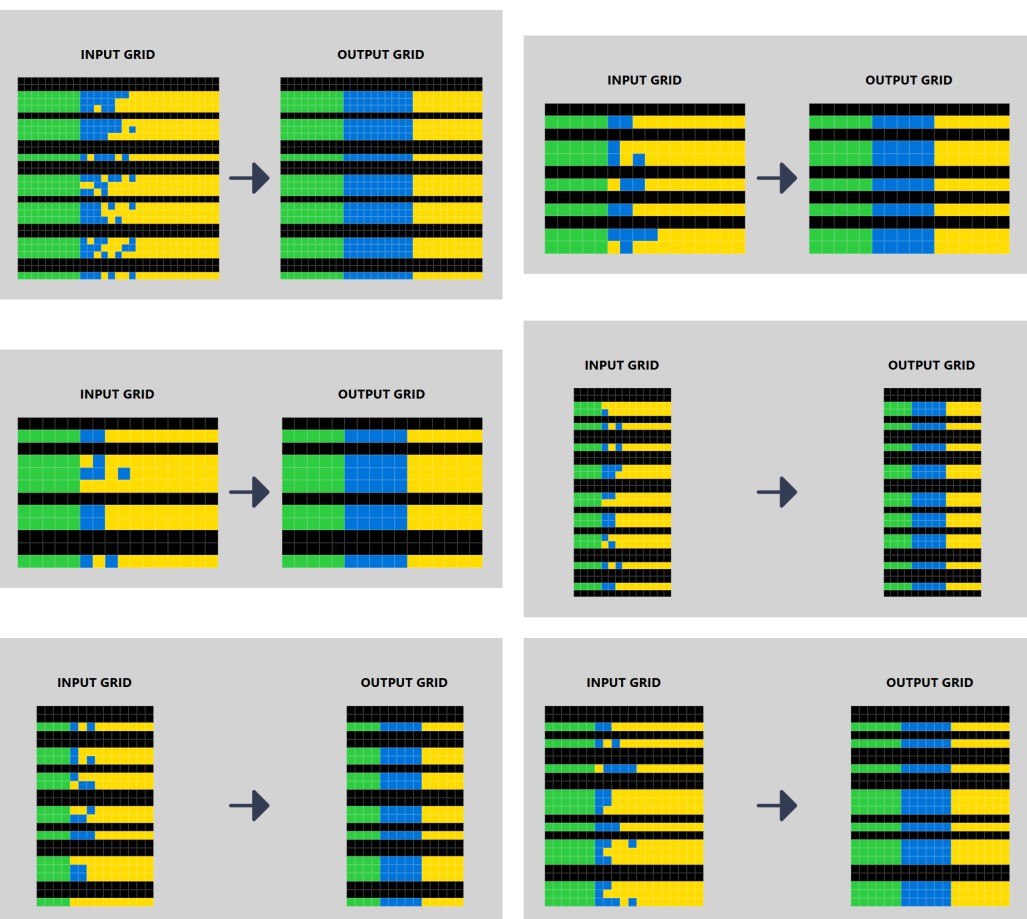

Figure 17: GIFARC-generated task 410.

E.10 EXAMPLE TASK OF TYPE 10 - WAVE & DIAGONAL FLOW

• Concepts : cyclical pulsing, polygonal wave propagation, time-lapse transformation

• Description : In the input, you will receive a sequence of frames representing a dark polygonal surface with green glowing points at each polygon vertex. Each frame gradually shifts the brightness of the polygons in wave-like motions, causing shimmering highlights that travel along the connected polygon faces. The initial frame and the final frame share the same luminous intensity, forming a seamless loop of pulsing and returning. Your task: 1. Identify the green points at the vertices, which remain spatially fixed throughout the sequence. 2. Track the wave-like shimmering across the polygon faces. 3. Ensure that the final frame's brightness pattern matches the initial frame's, capturing the cyclical nature of the pulse. 4. Output the entire transformation as a collection of frames (or a metadata structure) that visually loops back to the start. The key principles are: - The polygon mesh arrangement does not change; only the luminous intensity on the polygon faces and green vertices fluctuates. - Shimmering waves spread across the surface repeatedly, then recede, creating a looping timeline. - The result is a time-lapse style output in which the final state seamlessly resets to the first state, preserving the thought of continuous pulsing.

• Full web view is available at `https://gifarc.vercel.app/task/1354`.

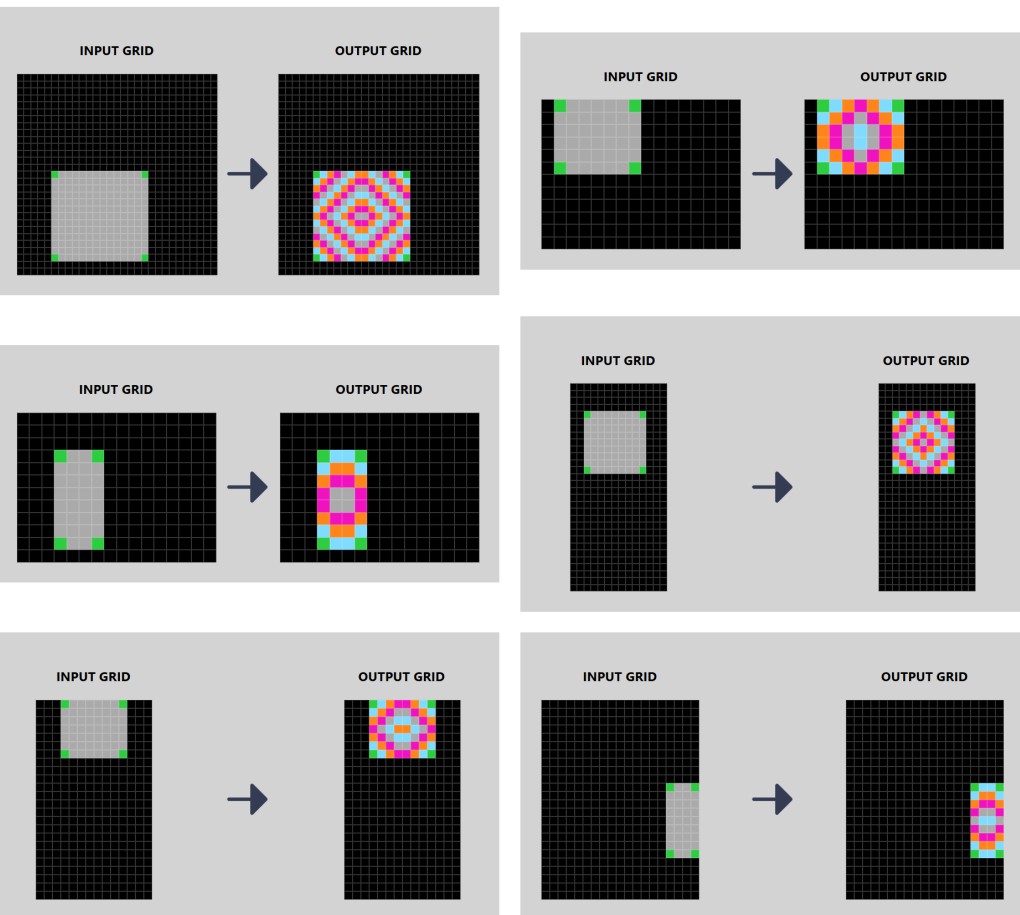

Figure 18: GIFARC-generated task 1354.

## E.11 EXAMPLE TASK OF TYPE 11 - GRAVITY & LIQUID FLOW

- Concepts : time-lapse, incremental changes, layering, chart visualization

- Description : In the input you will see multiple "frames" of a chart with vertical bars and a pink line, each frame on a separate grid. The black background, rectangular grid, and bounding box remain constant across these frames. In each subsequent frame, the vertical bars shift in height and flicker in intensity, while the pink line oscillates up and down with a net upward trend.

- Full web view is available at `https://gifarc.vercel.app/task/1413`.

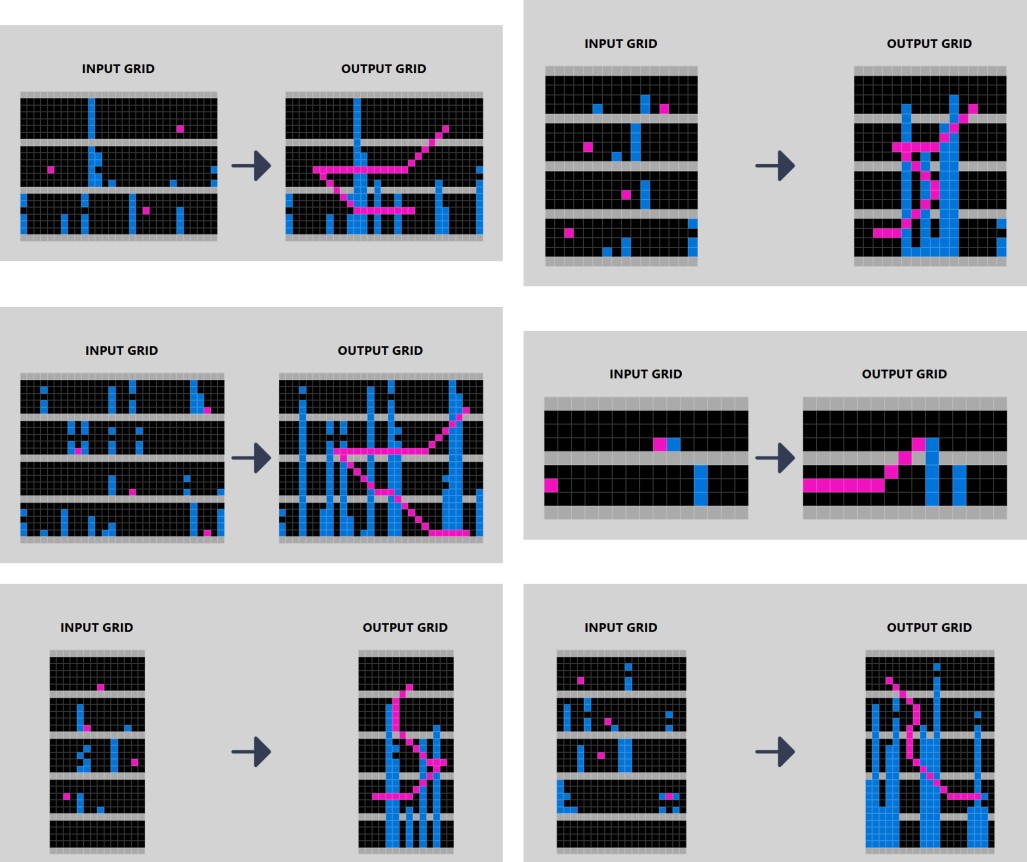

Figure 19: GIFARC-generated task 1413.

### E.12 Example Task of Type 12 - Slow Environmental Change

- Concepts : time-lapse, incremental changes, layering, chart visualization
- Description : In the input you will see multiple "frames" of a chart with vertical bars and a pink line, each frame on a separate grid. The black background, rectangular grid, and bounding box remain constant across these frames. In each subsequent frame, the vertical bars shift in height and flicker in intensity, while the pink line oscillates up and down with a net upward trend.
- Full web view is available at `https://gifarc.vercel.app/task/1554`.

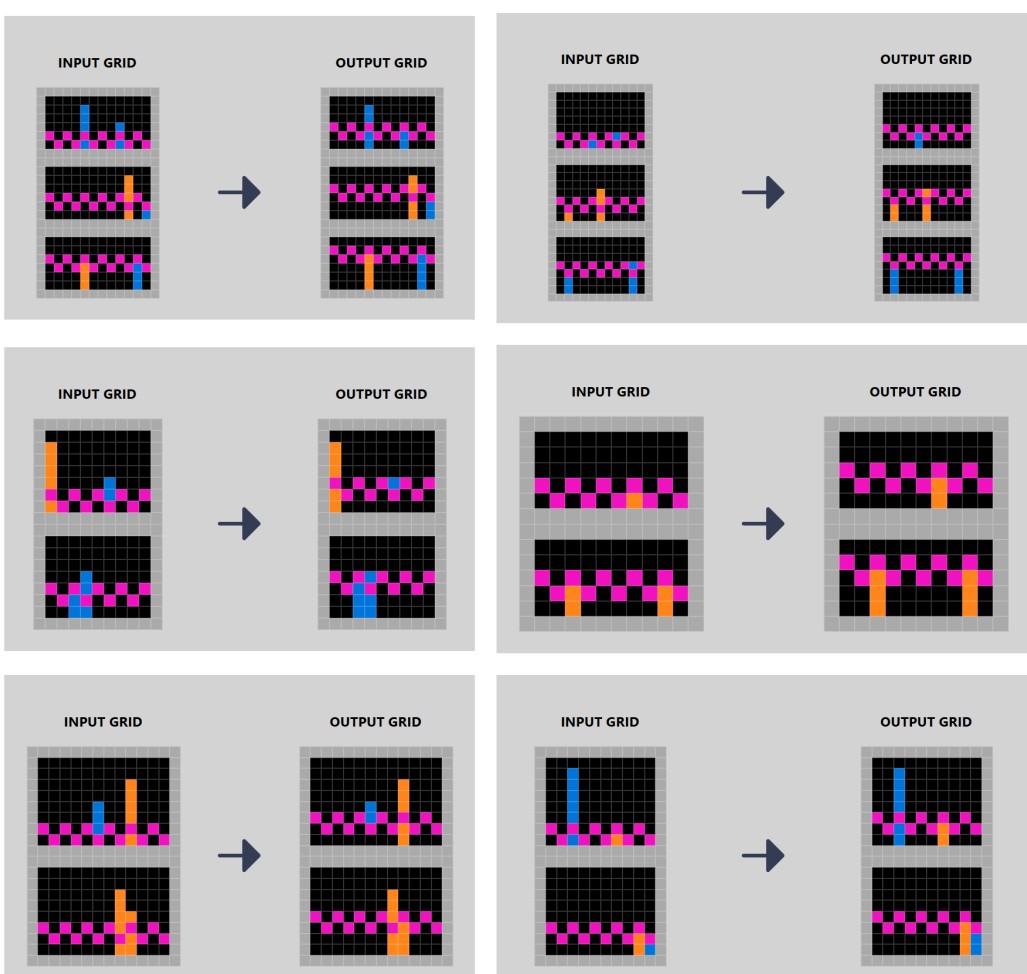

Figure 20: GIFARC-generated task 1554.

### E.13 EXAMPLE TASK OF TYPE 13 - SCALING BURST & SHAPE MORPHING

- Concepts : repetitive scaling, cyclical emergence, timed transformations

- Description : In the input you will see a pink background with one or more black rings near the center. Each ring should continuously expand outward from the center until it goes off the grid, at which point it disappears. Meanwhile, whenever an existing ring crosses halfway toward the border, a new smaller ring is spawned in the center. This process loops indefinitely—old rings vanish at the boundary, and new rings keep forming in the center. The output should capture this entire cycle, from the initial state to the eventual large rings that disappear, and newly formed rings that repeat the pattern. In a static puzzle context, illustrate at least one full cycle of rings moving outward and disappearing, while new rings appear at the center.

- Full web view is available at https://gifarc.vercel.app/task/651.

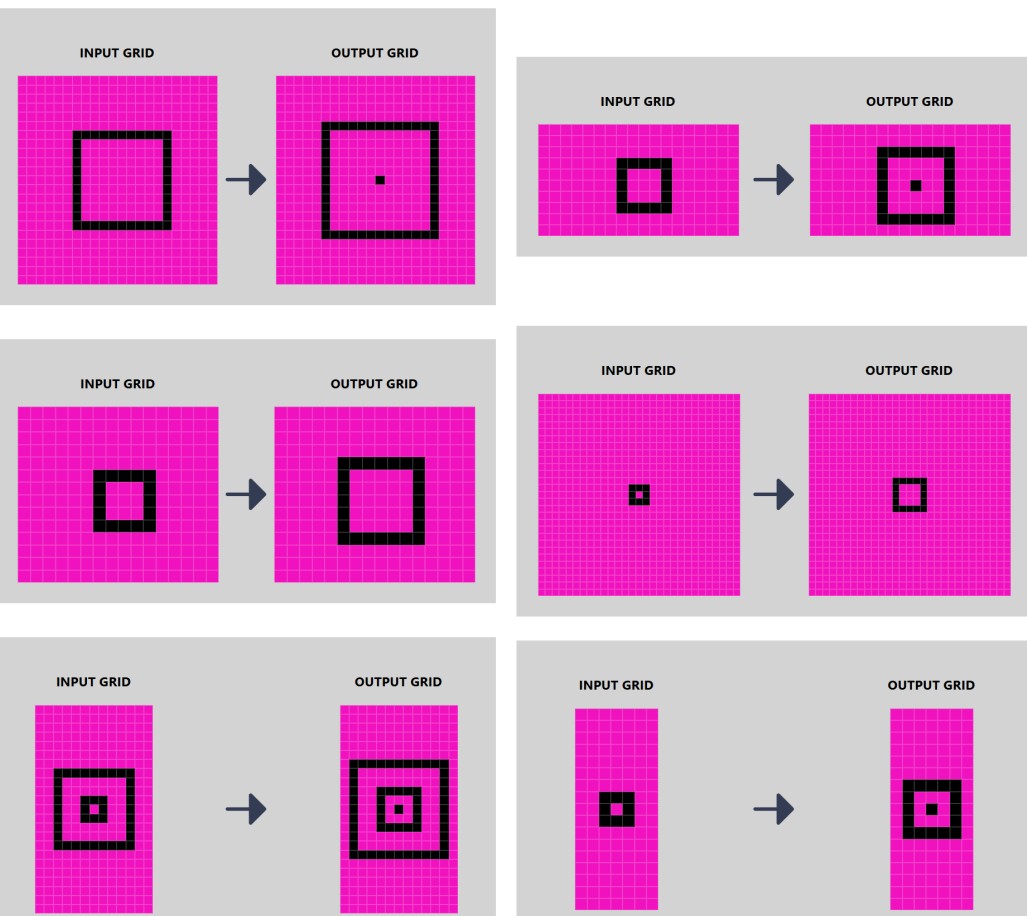

Figure 21: GIFARC-generated task 651.

### E.14 EXAMPLE TASK OF TYPE 14 - ATTACH/DETACH CLUSTERS

- Concepts : multi-frame transformation, attachment/detachment, motion, object grouping

- Description : The input represents a time-sequence (e.g., multiple "frames") showing a suited host seated at a desk with several static background objects (gift bag, city backdrop, building spire). The host reaches for a black mug, and we see that the tie is inadvertently attached to the mug (one frame shows the tie moving upward with the mug). In subsequent frames, the host detaches the tie from the mug and returns both to their normal positions. The core principle to illustrate is that if two objects are "attached" they move together until they become detached. Your output is the final frame where the tie is no longer stuck to the mug. The puzzle solution must show how the tie becomes free again, returning to its intended resting position, while the mug is back on the desk.

- Full web view is available at https://gifarc.vercel.app/task/7128.

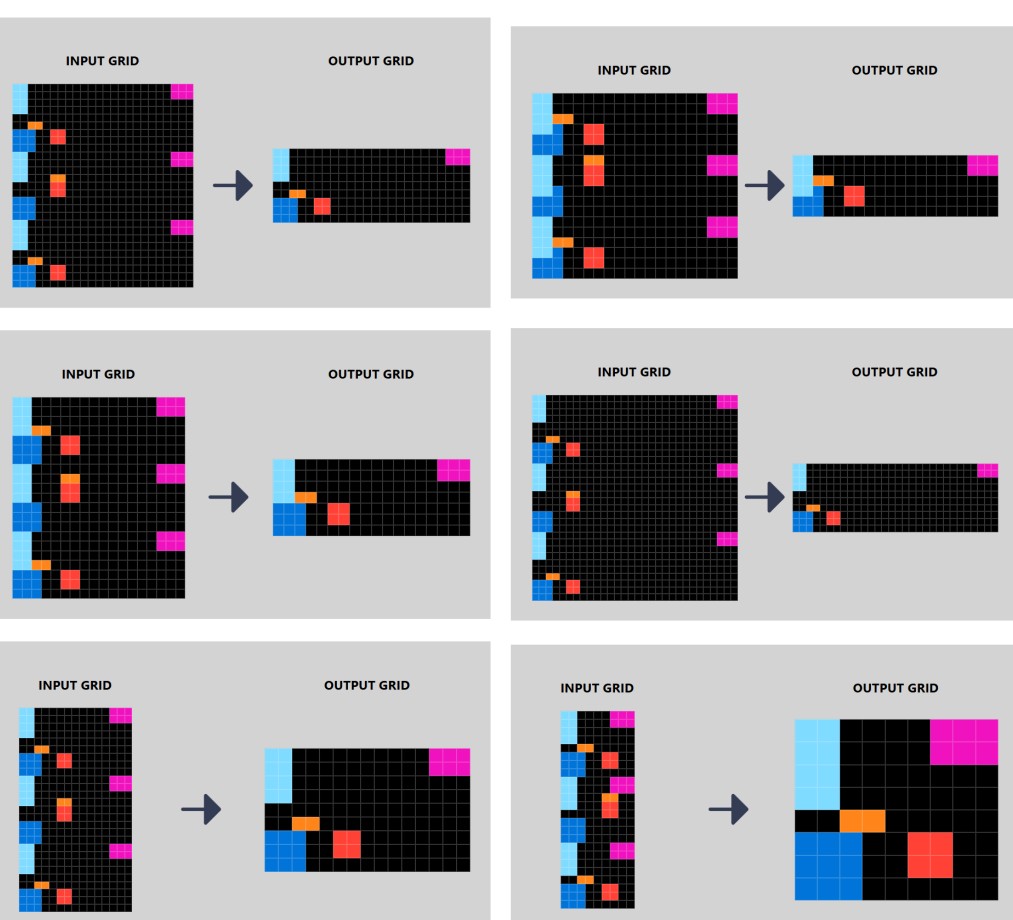

Figure 22: GIFARC-generated task 7128.

## E.15  EXAMPLE TASK OF TYPE 15 - LAYER SEPARATION & MERGING

- Concepts : color gradient, layered squares, temporal progression, partial transparency

- Description : In the input you will see a multi-frame sequence of an overlapping grid of squares starting in dark blues at the top, transitioning gradually to lighter, more yellow tones at the bottom. The squares remain in a consistent arrangement, but their colors shift slightly from frame to frame. Overlaps create partial transparency effects that slightly modify the underlying tones. To produce the output, you must show the final frame in which the color gradient transitions all the way to lighter shades near the bottom. Capture the entire transformation by preserving the arrangement of squares, ensuring that no abrupt color changes occur between consecutive frames, and maintaining a seamless progression of hues through the final arrangement.

- Full web view is available at `https://gifarc.vercel.app/task/214`.

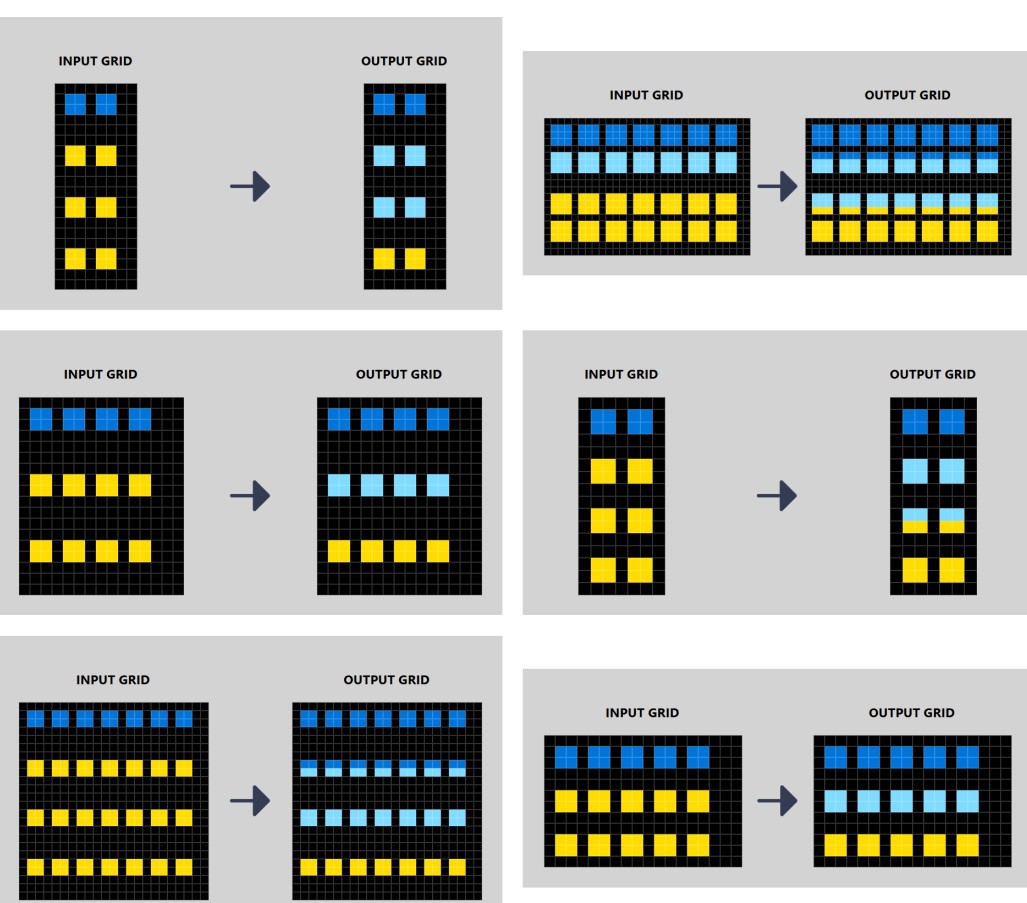

Figure 23: GIFARC-generated task 214.

### E.16 EXAMPLE TASK OF TYPE 16 - TEXT & PUNCTUATION TRANSFORMATION

- Concepts : reflection, wave distortion, layering

- Description : In the input grid, you will see: 1. A forest background occupying the top portion. 2. The word "RELAX" in large, two-toned letters at the center of the grid. 3. A black area (representing water) at the bottom. To make the output: 1. Reflect both the forest background and the "RELAX" text onto the black water region, mirroring them vertically. 2. Apply a horizontal wave distortion row by row to the mirrored content, simulating water ripples. 3. Preserve the original forest and text unchanged in their original positions, keeping them static above the water. The core principle is to create a continuous rippling motion in the lower part that reflects the static background and text above.

- Full web view is available at https://gifarc.vercel.app/task/3896.

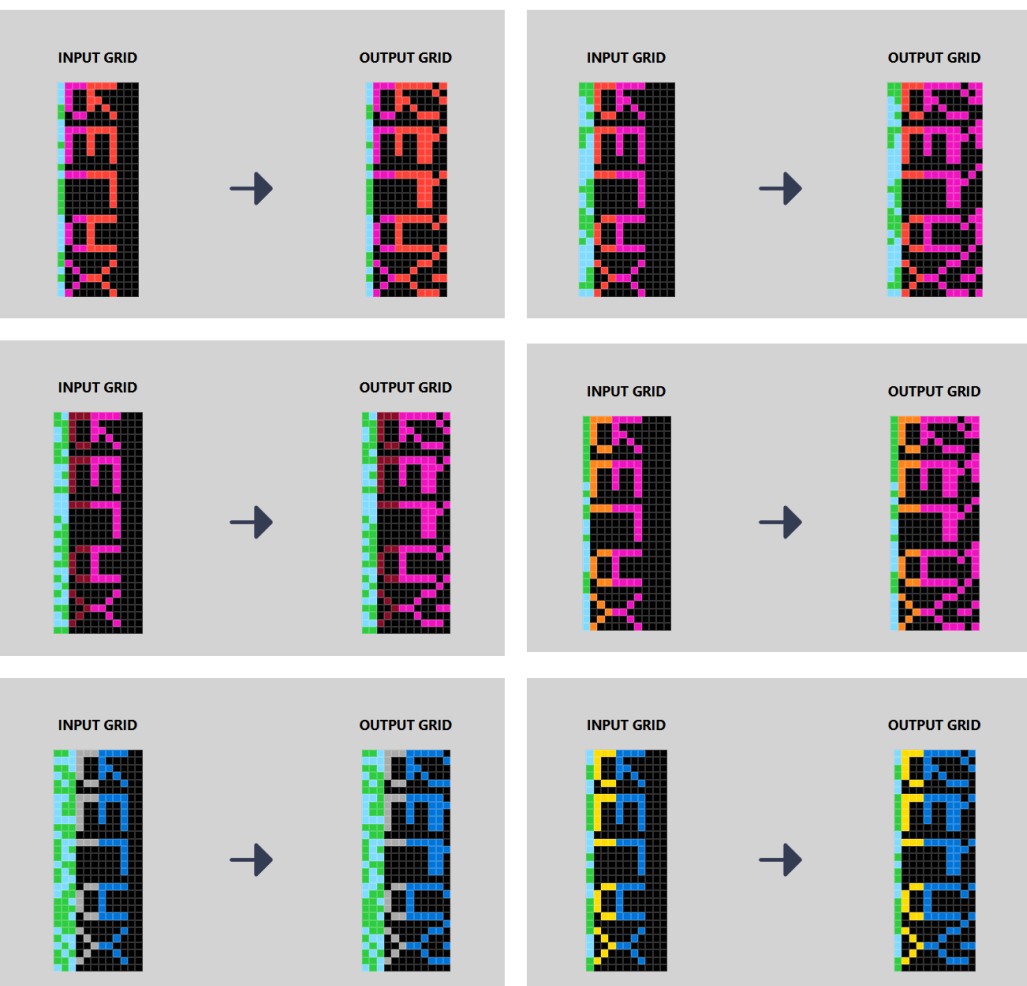

Figure 24: GIFARC-generated task 3896.

### E.17 Example Task of Type 17 - Minimal Motion Overlay

- Concepts : ring arrangement, sequential flipping, cyclical motion

- Description : In the input you will see a grid composed of multiple rows (or columns), each representing a separate "frame" of a green cross-like shape (with a purple outline) on a blue background. 1. The shape is supposed to expand and contract in a symmetrical, cyclical pattern around its center, but the given frames may be scrambled or have missing corners/edges. 2. Reorder these frames (and fix any missing corners/edges) to restore a clean, cyclical sequence where the cross-like shape starts in its initial state, morphs outward, and returns symmetrically to a state matching the initial frame. 3. The shape should retain its green fill, purple outline, and be fully contained in the blue background across all frames. 4. The final output is a single grid (or list of frames) showing the corrected sequence in order, with all frames centered and fully symmetrical, completing one full expansion-contraction cycle.

- Full web view is available at https://gifarc.vercel.app/task/9842.

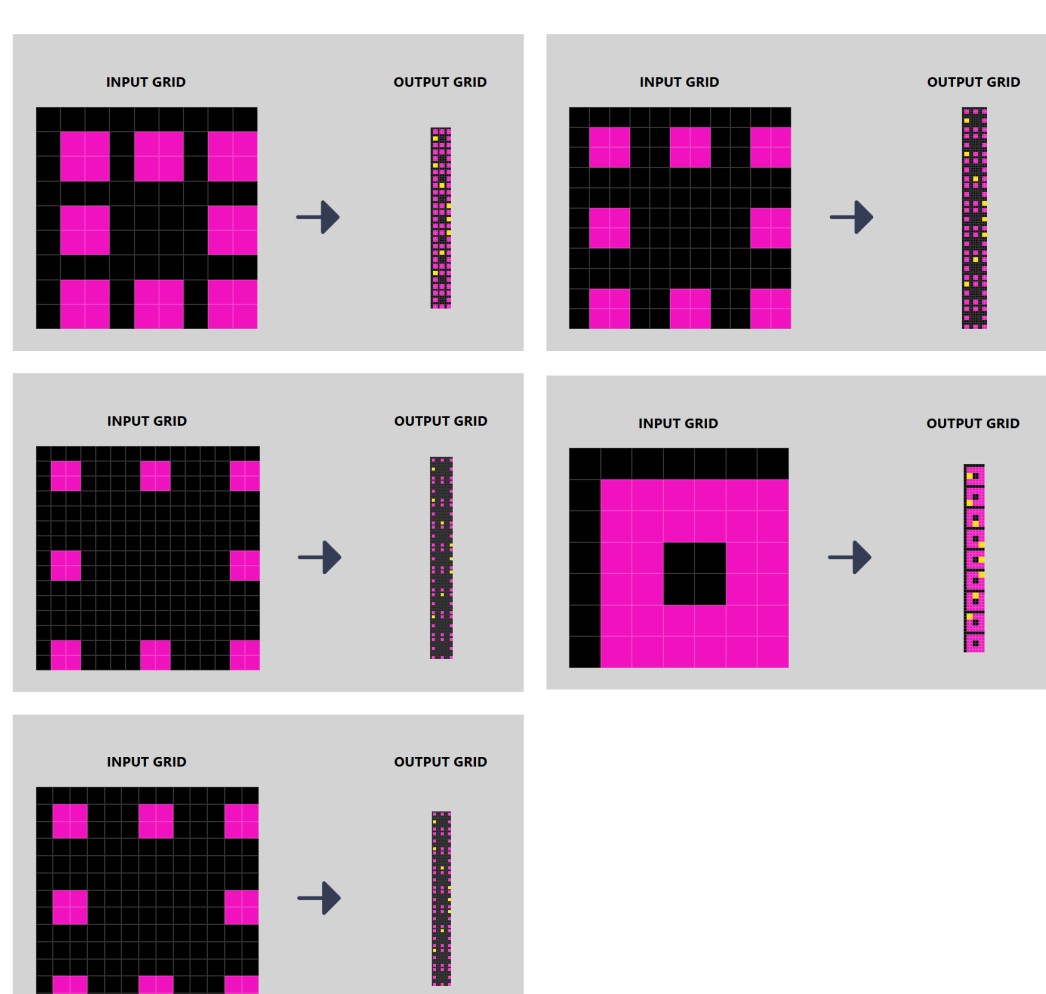

Figure 25: GIFARC-generated task 9842.

### E.18 EXAMPLE TASK OF TYPE 18 - STATIC VERIFICATION & NO CHANGE

- Concepts : multi-frame static verification, scene consistency, no-change detection

- Description : In the input, you will see several frames that together depict a stylized scene: - A top sphere positioned at the upper center - A curved band placed directly below the sphere - Two large symmetrical side shapes in yellow - A vertical pink line in the center - A signature text in the bottom right corner These elements are repeated exactly across all frames with no movement or color transitions. To make the output, you must check if all frames are truly identical. If they are, return a single-frame grid replicating the scene exactly. If there is any discrepancy (in shape, color, or position) among the frames, then the output should be a black canvas of the same size, indicating the scenes are not perfectly static.

- Full web view is available at https://gifarc.vercel.app/task/413.

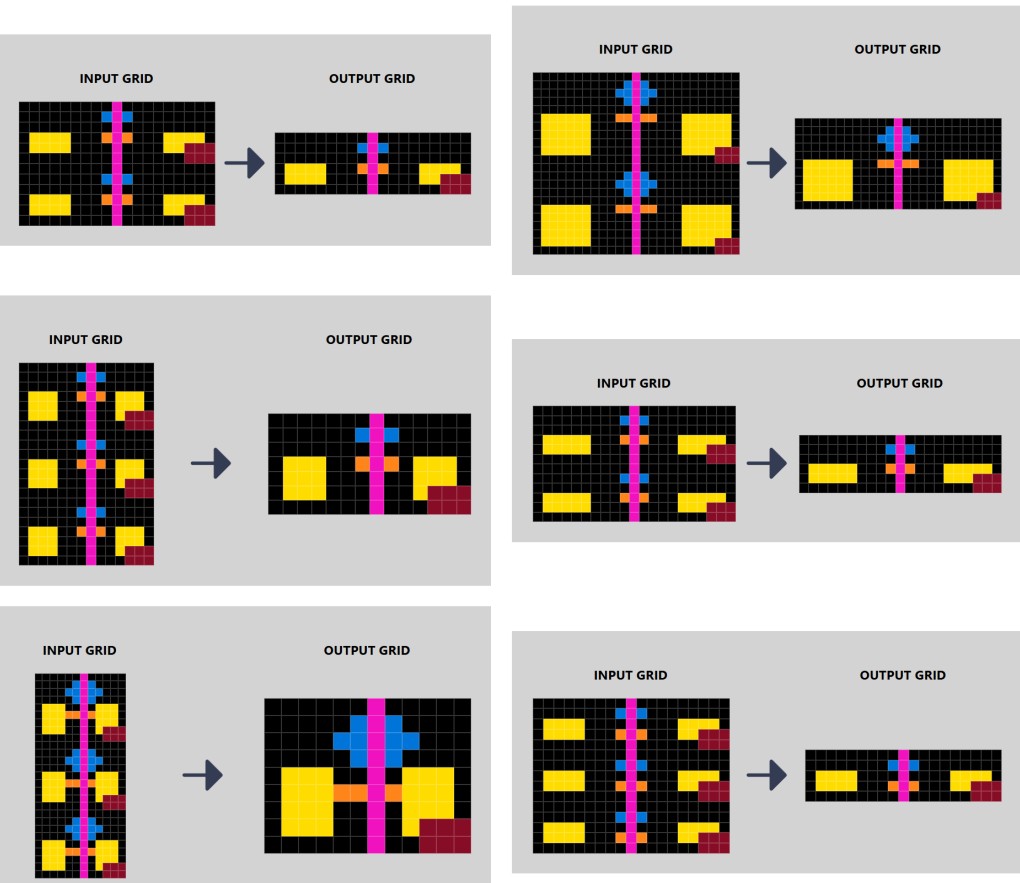

Figure 26: GIFARC-generated task 413.

## E.19 EXAMPLE TASK OF TYPE 19 - FRACTAL EXPANSION & SELF-SIMILAR REPEATS

- Concepts : fractal expansions, symmetrical pulsations, radial transformations, iterative growth

- Description : In the input, you will see a sequence of grids (frames) on a black background. Each grid depicts: - A large circular region in the center, with multiple concentric rings and radial lines diverging outward. - A stacked-curve fractal anchored at the bottom-left corner. - A set of spire fractals anchored at the bottom-right corner. As you move from one frame to the next in the input, these elements undergo iterative transformations: - The radial lines repeatedly shift in thickness and visual intensity, while preserving rotational symmetry. - The concentric rings in the circular region pulsate by alternately expanding and contracting. - The stacked-curve fractal on the bottom-left changes its curve density, fractally adding or removing segments. - The right-side spire fractals expand and contract in repeated vertical segments. Your task is to replicate and apply these transformations for the entire sequence, and produce the final frame of the animation as the output: 1) Ensure the anchoring of the fractals at the bottom edge remains the same. 2) Preserve the radial symmetry of the central circular region and its common center point. 3) For each pulsation step in the input, magnify or contract the rings, lines, and fractals accordingly. 4) Continue until the final pulsation step is reached. That final state is your output grid. The essential principle is that symmetry-based fractal transformations and repeated cyclical expansions/contractions produce the overall pulsating effect. By reflecting each iterative change step by step, you reconstruct the final pulsating pattern visible at the end of the sequence.

- Full web view is available at https://gifarc.vercel.app/task/155.

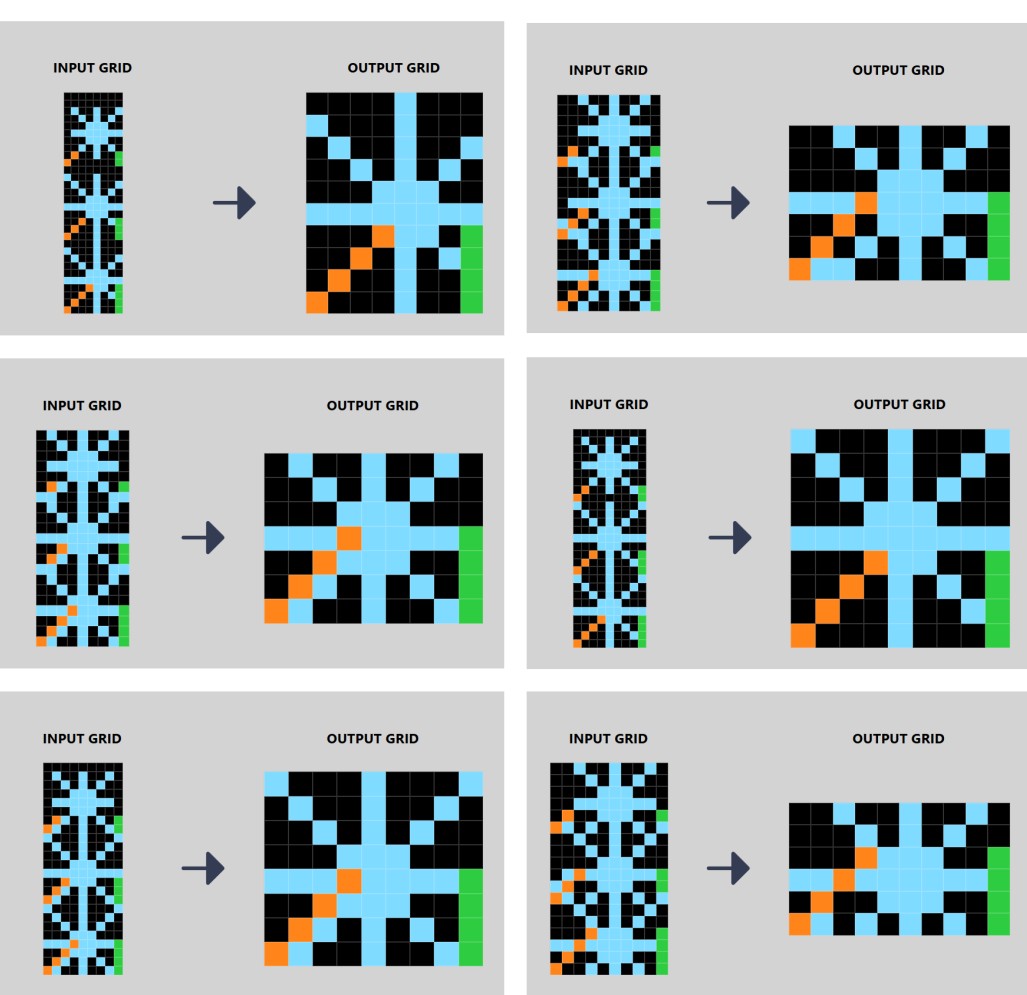

Figure 27: GIFARC-generated task 155.

## E.20 EXAMPLE TASK OF TYPE 20 - SEQUENTIAL PATTERN GROWTH & TRANSITION

- Concepts : fractal expansions, symmetrical pulsations, radial transformations, iterative growth

- Description : In the input, you will see a sequence of grids (frames) on a black background. Each grid depicts: - A large circular region in the center, with multiple concentric rings and radial lines diverging outward. - A stacked-curve fractal anchored at the bottom-left corner. - A set of spire fractals anchored at the bottom-right corner. As you move from one frame to the next in the input, these elements undergo iterative transformations: - The radial lines repeatedly shift in thickness and visual intensity, while preserving rotational symmetry. - The concentric rings in the circular region pulsate by alternately expanding and contracting. - The stacked-curve fractal on the bottom-left changes its curve density, fractally adding or removing segments. - The right-side spire fractals expand and contract in repeated vertical segments. Your task is to replicate and apply these transformations for the entire sequence, and produce the final frame of the animation as the output: 1) Ensure the anchoring of the fractals at the bottom edge remains the same. 2) Preserve the radial symmetry of the central circular region and its common center point. 3) For each pulsation step in the input, magnify or contract the rings, lines, and fractals accordingly. 4) Continue until the final pulsation step is reached. That final state is your output grid. The essential principle is that symmetry-based fractal transformations and repeated cyclical expansions/contractions produce the overall pulsating effect. By reflecting each iterative change step by step, you reconstruct the final pulsating pattern visible at the end of the sequence.

- Full web view is available at https://gifarc.vercel.app/task/2061.

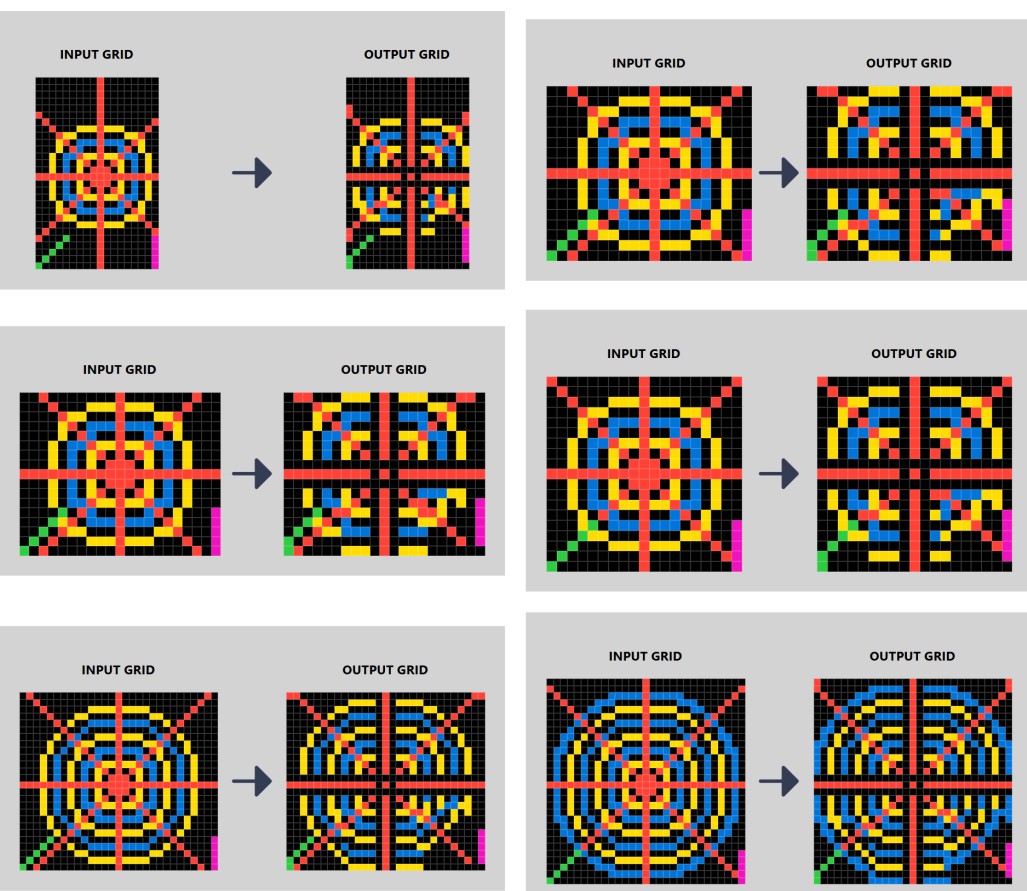

Figure 28: GIFARC-generated task 2061.

