# OpenReview forum: "GIFARC: Synthetic Dataset for Leveraging Human-Intuitive Analogies to Elevate AI Reasoning"
_ICLR.cc/2026/Conference — Submitted to ICLR 2026_

### Official Review · Reviewer_Cdhh · 2025-10-29

**Soundness:** 2
**Presentation:** 3
**Contribution:** 2
**Rating:** 4
**Confidence:** 4

**Summary:**

The paper proposes GIFARC, a synthetic dataset of ~10,000 ARC-style tasks generated from short, motion-rich GIFs to inject human-intuitive analogical structure into ARC problem solving. A three-stage, VLM-driven pipeline (visual abstraction → task sketch → ARC task with executable Python solution and an explicit analogy label) converts each GIF into input–output grid pairs, code, and a named analogy (e.g., “blocked water flow”). The authors evaluate along three dimensions: (i) in-context learning, where examples from GIFARC encourage models to describe tasks using human-aligned analogies; (ii) human/LLM judgments of analogy alignment; and (iii) supervised fine-tuning (SFT) on GIFARC, which substantially lowers NLL on ARC-AGI-1/2 but yields only 2.9% accuracy on ARC-AGI-1 (from a 0.2% baseline) and 0% on ARC-AGI-2. The dataset and prompts are released publicly; the pipeline itself relies on a proprietary VLM (“o4‑mini”), and later-stage generation exhibits ~19–20% failure rates requiring retries.

**Strengths:**

- The paper targets a widely acknowledged gap between human- and model-level generalization on ARC and argues that analogical grounding can constrain search and make reasoning more human-like—implemented here by mining analogies from real-world GIFs. This is a creative way beyond rule-only synthetic generators.

- GIFARC provides 10k tasks with executable solutions and explicit analogy labels, plus extensive appendices (prompts, statistics, visual examples) and links to data/visualization, enhancing reuse and analysis by the community.

- Experiments show that “flattening” out analogy cues degrades models’ abstract descriptions, while human/LLM judgments confirm the analogical labels are meaningfully aligned with human concepts.

- The paper provides effective figures and qualitative failure analyses that highlight where analogy identification and execution can be decoupled.

**Weaknesses:**

- Despite reductions in negative log-likelihood (NLL), exact-match accuracy improves only to 2.9% on ARC-AGI-1 and remains at 0% on ARC-AGI-2—well below recent ARC benchmarks, including the ARC Prize-winning results. This undercuts the core claim of “bridging the gap” between models and human-level reasoning.

- The main experiments employ supervised fine-tuning (SFT) on direct grid prediction, whereas state-of-the-art ARC solvers use program synthesis or neuro-symbolic pipelines with test-time search. This mismatch makes it difficult to assess GIFARC’s true value as a source of priors or analogical guidance for code/search-based frameworks.

- The pipeline’s heavy reliance on the closed/proprietary VLM “o4-mini” severely limits reproducibility, cost transparency, and future stability. The paper reports ~19–20% failure rates in later stages, raising concerns about the scalability and robustness of the data generation process.

- Many ARC tasks hinge on abstract, symbolic, or topological relations not easily expressed or extracted from GIF-based analogies (such as symmetry, flow, or discrete logical rules). Experiments indicate cases where analogy detection succeeds but execution fails, highlighting representational limitations of the pipeline.

- All reported fine-tuning results center on a single model (Mistral-NeMo-Minitron-8B-Base) trained for a fixed number of epochs, with no discussion of a hyperparameter search.

**Questions:**

- Given the significant NLL reductions but only 2.9% / 0% accuracies on ARC-AGI-1/2, what concrete bottlenecks prevent improved likelihoods from translating into exact-match solutions?

- Have you substituted “o4-mini” with open-source VLMs in any pipeline stages, and can you ablate components (with/without analogy labels, sketches-only, code-only) to pinpoint what drives the observed NLL improvements?

- Can you quantify the most common causes of the reported ~19–20% stage failures, number of retries, and the overall API, time, and monetary generation costs?

- How do the analogy categories in GIFARC compare to the latent concept space of ARC? Do GIFs bias toward physical and dynamic analogies, and are there observed systematic blind spots for more abstract/symbolic relations?

- How do humans perform on GIFARC tasks, versus the original ARC distribution? Does difficulty match, and are common error types similar or different?

---

> ### Author Response · Authors · 2025-11-23
>
> Thank you very much for your valuable feedback and suggestions.
>
> ---
>
> ## [Q1] Bottlenecks preventing NLL reductions from translating into exact-match accuracy
>
> As you insightfully noted regarding the weaknesses, SFT may not be the optimal approach to fully leverage the value of GIFARC; consequently, we believe test-time search is essential to overcome these limitations. While SFT-based direct grid prediction is vulnerable to error accumulation due to its reliance on a single forward pass, test-time search methods can mitigate these errors by generating multiple candidates and iteratively refining them. To address the limitations of SFT (e.g., error accumulation), we plan to incorporate evolutionary test-time compute [1], using GIFARC’s analogies to guide candidate refinement. Also, we will apply GIFARC to the latest ARC solver papers including TRM[2].
>
> To further validate our current SFT approach, we are currently conducting an ablation study that compares (1) Full components (Analogy+Code+Grid), (2) w/o Code, (3) w/o Analogy, and (4) Baseline (Grid only). Once this experiment is completed, we will analyze which factors drive the NLL improvements, isolate the contribution of each training component, and incorporate the results into both our follow-up response and the revised manuscript.
>
> ---
>
> ## [Q2] Generation Pipeline: Open-Source Model and Failure Analysis
>
> Regarding the generation pipeline, we prioritized o4-mini for its multimodal performance but acknowledge the customizability benefits of open-source models for future work. Our analysis of the GIFARC generation pipeline identified two primary failure causes, which are content policy violations triggered by the API and LLM outputs that failed format validation. We will include a detailed discussion of these failure modes in the revision to provide insights into the operational challenges.
>
> ---
>
> ## [Q3] Comparison of analogy categories and systematic blind spots
>
> Thank you for pointing out the potential biases in GIFARC's analogy coverage. To compare GIFARC's analogy categories with ARC, we categorized tasks from the ARC-AGI-1 training and evaluation sets using the 20 category tags presented in Appendix A of our paper. Since there is no official taxonomy from the ARC-AGI team, we used our proposed 20 category tags to conduct a comparative analysis of the categorical distributions between GIFARC and ARC-AGI-1. The analysis results show that in ARC-AGI-1, Layer Separation & Merging accounted for the highest proportion at 11.3%, followed by Gravity & Liquid Flow at 9.6% , with an average of 36 tasks per category and a standard deviation of 26. GIFARC covers 90.7% of the categories found in ARC-AGI-1 training and evaluation, and can supplement task types not present in the original training set. The detailed structured descriptions have been added at the “Comparing Taxonomy Distributions“ of Appendix A.
>
> However, we acknowledge that abstract and symbolic relations unsuitable for GIF visualization are underrepresented in GIFARC, particularly tasks involving discrete logical operations. This reflects a fundamental trade-off where GIFs excel at encoding continuous visual transformations but struggle with abstract meanings. Importantly, this scope aligns with our specific design goal because our research focused on evaluating analogical reasoning capabilities for physical and dynamic transformation relations which constitute an essential subset of ARC tasks. We will explicitly clarify this intended scope and the distinction from purely symbolic reasoning in the revised manuscript.
>
> ---
>
> ## [Q4] Human performance and error types
>
> We have not yet benchmarked human solving performance on GIFARC. We agree that the "representational gap" between dynamic GIFs and static grids warrants investigation, as discrepancies between analogy detection and execution can arise.  In future work, we will employ not only LLM-based evaluation but also human evaluator-based approaches to mitigate this representational gap.
>
> **References:**
>
> [1] Berman, J. (2025). *How I got the highest score on ARC-AGI again*. Substack. Available at: https://jeremyberman.substack.com/p/how-i-got-the-highest-score-on-arc-agi-again
>
> [2] Jolicoeur-Martineau, Alexia. "Less is More: Recursive Reasoning with Tiny Networks." arXiv preprint arXiv:2510.04871 (2025).

---

### Official Review · Reviewer_NRMS · 2025-10-31

**Soundness:** 2
**Presentation:** 1
**Contribution:** 2
**Rating:** 2
**Confidence:** 4

**Summary:**

The authors use VLMs to synthesize (in 3 different stages) a new dataset, called GIFARC, of ARC-style tasks that are "inspired" by GIFs from an online collection.  These ARC-like tasks are annotated with "analogy labels" that are generated by the VLMs.

In one experiment, the authors gave GPT 4.1-mini in-context examples from GIFARC, and then asked it to solve tasks from ARC-AGI-2.  They experimented with three types of in-context examples:

-- 15 examples from GIFARC dataset with full description

 -- 15 examples from GIFARC where researchers have replaced "analogic terms" by "lower-level synonyms"

 -- 15 examples from GIFARC where researchers have replaced "analogic terms" by "lower-level synonyms" and the "solution" (Python program encoding the transformation) is not given

In a second experiment, a pre-trained LLM fine-tuned on examples from GIFARC, and then it is used to solve tasks from ARC-AGI-1 and ARC-AGI-2 evaluation set.  There is marginal improvement on ARC-AGI-1.  No tasks from ARC-AGI-2 are solved.



In a second

**Strengths:**

The paper attempts to incorporate analogical reasoning into a method for improving LLM ARC solvers, in an original way.

**Weaknesses:**

There are two main weaknesses:

First, it was a struggle to understand this paper---there are many aspects of it I found unclear.  It would really help to have a running example to illustrate the steps of starting with a GIF and generating a GIFARC task.   It would also be really helpful to have examples for the "full-description", "without analogy", and "without analogy and solution" data.    Also, Section 4.2 was particularly unclear.

Second, the authors make claims that are not clearly supported by the evidence given in the paper, and the paper lacks methods for evaluating those claims.   First, it's not clear how successful VLMs are at generating useful "analogy-laden" descriptions and then tasks.   Second, it's not clear how much the in-context examples enable the LLM to generate a correct / useful analogy.  Figure 3 gives one example where a correct analogy is given, but this is just one example.  It would be useful to give results on the percentage of tasks that have a correct analogy and correct output grid, correct analogy and incorrect output grid, incorrect analogy and correct output grid, etc.
Finally, for the fine-tuning experiment, the hypothesis seems to be that fine-tuning on GIFARC is what is causing the improvement over the base model.  It would be good to do a control experiment -- would an improvement over the base model still be seen if the fine-tuning data was, say, the ARC-AGI training set instea of GIFARC?  The authors need to show in a more compelling way that GIFARC is actually doing something above and beyond simple fine-tuning on ARC-like tasks.

**Questions:**

The authors state: ""Despite substantial progress in deep learning, state-of-the-art models still achieve accuracy rates of merely 40-55% on the 2024 ARC Competition" -- this is technically true, since o3-preview didn't compete, but o3-preview in high effort mode got 88% on semi-private test set.  It is misleading to leave this out.

Figure 1: "Illustration of two different solutions of ARC-style task found with or without analogic approach.  -- Found by who or what?
Also, how representative is this of the overall results?

"Our first stage converts a raw GIF into a structured, readable summary that captures the analogy-laden visual logic hidden in the clip." -- what do you mean by "analogy-laden visual logic"?  What's an example?  When I look at these examples in the appendix, I don't see a lot of what I would consider to be analogies.

"The final stage transforms each task sketch produced in Step 2 into a fully executable ARC-style task."  What does "fully executable" mean here?  Are these tasks actually solvable in some human-like way?

Section 4.2 "Next, GPT 4.1-mini with full description and GPT 4.1-mini with analogy-removed description were instructed to find the analogy implied in the 12 ARC-style tasks."  Is there a prompt given in the appendix for this? And which 12 ARC-style tasks are you talking about?

"We first instructed o3-mini to evaluate semantic similarities by providing a one-shot guideline about measuring similarity." -- Please explain what you did in more detail.

Figure 4 is hard to understand.  What is being measured in the "Human", "Full Description", and "Analogy-Removed description bars? These are similarities between what and what?  What do the actual values mean?

"we embedded each outputs and compared the cosine similarity between" -- how did you do the embeddings?


The authors state, "We have now confirmed that the components are analogically meaningful." but how exactly has this been confirmed, and what does "analogically meaningful" mean?

Similarly, the authors state, "we confirmed that GIFARC provides analogical reasoning  guidance to models".  Again, this doesn't seem to be confirmed by the results given in the paper.  Is there a way to show this more quantitatively?

"NLL values indicate a higher likelihood of the model generating the correct answer" -- Do you mean "lower  NLL values" indicate this?

"the substantial NLL improvements across both evaluation sets demonstrate that the model is learning meaningful patterns and moving closer to correct solutions." -- How do you verify this?

---

> ### Author Response · Authors · 2025-11-23
>
> We appreciate your comments and suggestions.
>
> ---
>
> ## [Q1] **Clarifications on SOTA Comparisons and Figure 1**
>
> ---
>
> We acknowledge that the mention of SOTA models needs to be more comprehensive. While o3-preview demonstrates high accuracy on the ARC-AGI-1 semi-private test set, it relies on substantial computational resources. Our work focuses on a different axis of contribution, which is efficiently injecting analogical priors into agents to bridge the reasoning gap found in open-weights models. Regarding Figure 1, we clarify that this figure is a conceptual illustration designed to visualize the theoretical mechanism of analogical reasoning rather than an empirical result from our model. We have updated the caption in the revised manuscript to explicitly state that this is a conceptual diagram to prevent any misunderstanding.
>
> ## [Q2] On Step 1, 3 of GIFARC pipeline
>
> ---
>
> In our first stage, we employed a VLM to process the GIF into a text-based summary, which will be later used in building an analogy-laden visual logic in the second step. For instance, a GIF shown in [https://gifarc.vercel.app/v1.1/task/3704](https://gifarc.vercel.app/v1.1/task/3704?shuffle_seed=0) is processed in step 1. For example, for the task shown in the provided URL (task 3704), the VLM successfully extracted explicit objects (e.g., 'blue face', 'green leaves'), dynamic patterns ('cyclical bobbing'), and core principles ('synchronization').
>
> In the final stage, the pipeline generates a Python code snippet. We define "fully executable" as code that not only implies a solution but deterministically maps input grids to output grids via runnable Python functions.  To further ensure the quality and solvability of these tasks, we enforce 8 rigorous filtering conditions, including checks for non-determinism, color invariance violations, and identity mappings, as detailed in Appendix C.2.
>
> ## [Q3] On the first experiment (section 4.1)
>
> ---
>
> We define a component as “analogically meaningful” if it enhances the model's ability to describe the task using correct analogies. Our case study in Section 4.1 suggest preliminary evidence that providing richer GIFARC components leads to gradual improvements in the model's analogical explanations, thereby validating the contribution of each dataset component.
>
> ## [Q4] On the second experiment (section 4.2)
>
> ---
>
> In Section 4.2, we employed a dual-metric approach to evaluate the alignment between ground-truth analogies and those generated by models and humans. First, we utilized o3-mini to assign similarity scores (0–1) based on a one-shot prompt designed to assess whether the generated analogy captures the core concepts of the ground truth. Second, as illustrated in Figure 4, we measured the cosine similarity between the ground-truth analogies in the GIFARC dataset and the corresponding explanations generated by each agent and human solver group, using the text-embedding-ada-002 model.
>
> We emphasize that this experiment was designed as a high-fidelity exploratory investigation, prioritizing the precision of expert annotations over large-scale statistics. We focused on a curated subset of tasks to rigorously validate GIFARC’s fundamental analogical design. Due to space constraints in this rebuttal, the full prompts used for these evaluations are provided in the Appendix B.4 of the revised manuscript.
>
> ## [Q5] On NLLs and quantitative measures
>
> ---
>
> Quantifying a model’s “reasoning level” is inherently difficult, especially in our setting where a small open-weights model achieves very low exact-match accuracy on ARC-AGI. In the original submission, we reported NLL primarily as an auxiliary signal to see whether the model assigns more probability mass to the correct output grids after fine-tuning on GIFARC.
>
> However, we acknowledge that NLL has clear limitations as a metric for demonstrating the contribution of GIFARC. This is because it does not directly show whether a model trained on GIFARC is actually considering an appropriate analogy and solution strategy when solving a given ARC task. Therefore, in future experiments, rather than interpreting the value of GIFARC solely through aggregate performance measures such as accuracy and NLL, we plan to add analyses that explicitly examine which analogies the model generates to solve each task and how those analogies are used in its reasoning process.
>
> In addition, we are currently conducting a controlled comparison in which the model is trained only on ARC-AGI data, to examine how accuracy as the main metric and NLL as an auxiliary metric differ from the GIFARC setting. Going forward, we plan to further combine GIFARC with recent ARC solvers and investigate not only whether GIFARC improves their performance.

---

> ### Comment · Reviewer_NRMS · 2025-11-23
> **Reply to authors' rebuttal**
>
> Thank you for the detailed responses.  I hope that you will include these clarifications in the revised paper, add additional examples, and work on making the paper easier to understand.  (I still don't understand the meaning of "analogy-laden visual logic").    I am happy to hear that you are performing additional experiments to address some of my and other reviewers' concerns.  I think those will strengthen the paper, but of course the revised paper with the new results will need to be reviewed again, so I think that, given time constraints, the revised paper and re-reviewing will not be ready for ICLR this year.  Thus I will keep my assessment.

---

### Official Review · Reviewer_WL6M · 2025-11-02

**Soundness:** 2
**Presentation:** 2
**Contribution:** 1
**Rating:** 2
**Confidence:** 5

**Summary:**

This work proposes a synthetic data generation pipeline for ARC tasks that extracts analogy concepts from online GIFs. The dataset is used to fine-tune an LLM, showing limited improvement on ARC-AGI-1.

**Strengths:**

- The idea to use GIFs as a source of analogical transformations is interesting, and intuitively makes sense given that GIFs often contain visual motion, which is one of the important priors in ARC tasks.
- Synthetic data generation is a promising framework for improving abstract reasoning in LLMs.
- The fine-tuned model is evaluated on both ARC-AGI-1 and ARC-AGI-2.

**Weaknesses:**

- The primary weakness is that fine-tuning on the GIFARC dataset does not yield significant improvements. On ARC-AGI-1, the fine-tuning only improves performance from 0.2% to 2.9%, which is a very small improvement, and very poor performance both before and after fine-tuning. On ARC-AGI-2, performance is 0% both before and after fine-tuning. These results do not suggest that the dataset is successful at improving performance on ARC tasks.
- The LLM-evaluated similarity results shown in Figure 3 are not informative. First, the use of an LLM to evaluate similarity is not reliable. Second, similarity is not a useful metric here. It is possible for two descriptions to be similar (i.e., involve similar words of features) despite one being correct and one being completely wrong.
- There is not much discussion of where the GIFs that are used in data generation come from, and what kind of images they depict. Intuitively, it is not clear whether randomly sourced GIFs would actually be helpful in ARC tasks, so it would be good to include more discussion of the sorts of GIFs that are used and why they should be helpful for generating ARC tasks. Additionally, there is no validation that the generated tasks have anything to do with the GIFs that form the input to the pipeline. Do the generated tasks actually employ similar concepts as the input GIFs?
- There is no ablation provided for the specific importance of using GIFs. For instance, there should be a comparison with models fine-tuned on synthetic data generated using alternative pipelines, for instance using actual ARC tasks as input, or images instead of GIFs. The current results do not establish that GIFs are specifically useful above and beyond the potential benefit of synthetic data.
- The description of the state-of-the-art in ARC results is somewhat misleading. Although the best performing model that was officially entered in the ARC-AGI-1 competition achieved only 55% accuracy, other models that were not officially entered into the competition (due to using computational resources beyond those allowed in the competition) achieved performance very close to average human performance, most notably openAI's o3 model. ARC-AGI-2 results were worse, but overall the performance of state-of-the-art models is not as bad as is implied in the introduction.

**Questions:**

- Do the synthetically generated tasks meaningfully incorporate visual transformations extracted from the GIFs?
- How does the model perform when fine-tuned on data generated using alternative pipelines, including using ARC tasks or images as input?

---

> ### Author Response · Authors · 2025-11-23
>
> We appreciate your insightful comments and suggestions.
>
> ---
>
> ## [Q1] Do the synthetically generated tasks meaningfully incorporate visual transformations extracted from the GIFs?
>
> To address this question, we additionally conducted a new human evaluation on 1,000 randomly sampled GIFARC tasks.  To capture the core visual transformations, we extracted visual information in Step 1, and subsequently in Step 2, we extracted the concepts and descriptions necessary for task generation. The evaluation then focused on whether these concepts and descriptions were meaningfully incorporated into the final synthetic tasks.
>
> The result showed that 85.7% of the tasks demonstrated a proper alignment between the final tasks and their source concepts/descriptions. This result indicates that the majority of GIFARC tasks successfully integrate the dynamic visual elements from the concepts into the final task structure. The new result has been added to the “Qualitative Quality Evaluation” of Appendix A. Furthermore, to objectively assess dataset quality beyond human evaluation, we will develop a new quantitative method to measure how accurately the analogies extracted from GIFs are grounded in the generated GIFARC tasks.
>
> Additionally, as pointed out in the weaknesses about LLM similarity evaluation, the LLM-based evaluation in the Section 4.2 experiment is always exposed to the risk of hallucination. To address this, in future experiments, we will recruit 10 experts to conduct similarity evaluations of the analogies.
>
> ---
>
> ## [Q2] How does the model perform when fine-tuned on data generated using alternative pipelines, including using ARC tasks or images as input?
>
> We agree that it is essential to compare our GIFARC-based SFT model against models fine-tuned on alternative data sources such as the original ARC training set and other synthetic pipelines.
>
> To verify whether the current generation pipeline can produce high-quality datasets, we plan to compare it with other methodologies beyond the BARC generation pipeline that serves as the base of our current pipeline. Additionally, to confirm whether GIFs contain appropriate analogies for creating ARC tasks, we plan to conduct an ablation study comparing them with static media such as images. In this experiment, we intend to compare the performance of tasks created from images versus tasks generated from GIFs. This comparative experiment will measure how well models trained on each dataset (one from images, one from GIFs) solve evaluation tasks on ARC-AGI-1 and ARC-AGI-2.
>
> ---
>
> ## [Q3] GIFARC yields only marginal performance improvements
>
> We agree with the reviewer's concern that, although there is a non-trivial relative improvement from 0.2% to 2.9%, the absolute magnitude of this gain remains modest. To better establish GIFARC's contribution, we are currently conducting a controlled comparison to assess the effectiveness of GIFARC, in which we evaluate models trained separately on the ARC-AGI-1 training set and the ARC-AGI-2 training set. Once this experiment is completed, we will incorporate the results into an additional response and the revised manuscript.
>
> In addition, as another reviewer suggests, we believe that SFT alone may not be the most appropriate way to fully leverage the value of GIFARC. To address the limitations of SFT (e.g., error accumulation), we plan to incorporate evolutionary test-time compute [1], using GIFARC’s analogies to guide candidate refinement. Also, we will apply GIFARC to the latest ARC solver paper such as TRM[2]  which use open-source models.
>
> To further validate our current SFT approach, we are currently conducting an ablation study that compares (1) Full components (Analogy+Code+Grid), (2) w/o Code, (3) w/o Analogy, and (4) Baseline (Grid only). Once this experiment is completed, we will analyze which factors drive the NLL improvements, isolate the contribution of each training component, and incorporate the results into both our follow-up response and the revised manuscript.
>
> ---
>
> ## [Q4] Clarification on the State-of-the-Art ARC Performance
>
> We acknowledge that the mention of state-of-the-art models needs to be more comprehensive. Our work focuses on a different axis of contribution, which is efficiently injecting analogical priors into agents to bridge the reasoning gap found in open-weights models. We will revise the introduction in the manuscript to accurately reflect the current state-of-the-art, including both official competition results and subsequent high-performing systems. Reflecting this point, we have updated the introduction in the revised manuscript to incorporate these clarifications.
>
> **References:**
>
> [1] Berman, J. (2025). *How I got the highest score on ARC-AGI again*. Substack. Available at: https://jeremyberman.substack.com/p/how-i-got-the-highest-score-on-arc-agi-again
>
> [2] Jolicoeur-Martineau, Alexia. "Less is More: Recursive Reasoning with Tiny Networks." arXiv preprint arXiv:2510.04871 (2025).

---

> ### Comment · Reviewer_WL6M · 2025-11-26
>
> Thank you to the authors for these replies.
>
> For evaluating the link between the GIFs and the synthetically generated tasks, more detail is needed on the human evaluation method. It sounds like the authors asked participants to make a yes/no judgment about whether a task incorporated the visual transformations in a particular GIF. However this is not a particularly sensitive method, and may be biased by the implicit expectation for participants to answer affirmatively (i.e. it may be susceptible to demand characteristics). Additionally there is no detail about who the participants were. A better method would be to use a two-alternative forced choice paradigm, asking participants to select one of two GIFs that best matched the transformations in a task. It is also important to ensure that the participants are unbiased (i.e. they do not know the purpose of the study).
>
> Regarding the other proposals (e.g. additional baselines) these sound like good experiments to perform, but without these results their impact cannot be determined, and the primary concern of marginal performance gains remains. I think that more work is needed to validate this idea, but I encourage the authors to continue working on it.

---

### Official Review · Reviewer_A2YA · 2025-11-03

**Soundness:** 1
**Presentation:** 3
**Contribution:** 1
**Rating:** 2
**Confidence:** 4

**Summary:**

The paper introduces GIFARC -- an ARC-style dataset generated by leveraging GIF images to extract analogies, and then construct ARC-like problems with corresponding verbal descriptions. The authors run a number of experiments to validate the quality of their dataset.

**Strengths:**

-The authors investigate a highly relevant problem. The struggles of modern models with ARC-style tasks is indicative of a substantial gap in modern AI architectures or training approaches.
-The idea to extract visual analogies from GIF sources is original and promising.

**Weaknesses:**

While the idea of the paper is quite promising, I believe that there are very substantial flaws with experimental evaluation.

Some crucial aspects of evaluation are missing.

For example, it's crucial to check that the generated problems are human-solvable. The provided human evaluations are insufficient. Not only the sample is very small (three experts and 12 problems), but also analogy description is very different from solvability.

For example, while a human can, perhaps with some difficulty, find an analogy for the generated ARC-like problem shown on Figure 16, I believe nobody can be reasonably expected to be able to solve that task which essentially displays a medium-resolution pixel art of a flower/circular pattern with some nontrivial color transformation.

Many other tasks similarly appear either quite hard to solve or are, in my view, confusing. It's important to provide a reasonable human performance estimate for the dataset. In this scenario, the tasks need to be sampled at random.


Another issue is the abundance of unusual experimental design choices and lack of detailed procedure description.

For example, 4.1 (i) mentions that the tasks were selected after iterative refinement with LLM and human. But that makes these seected tasks non-representative of the general dataset which can not be refined in the same way (as that would be prohibitively time-consuming).

A related issue is the loose use of terms like "analogy" and even "solution", and lack of criteria for what constitutes a good analogy/solution. Figure 3 illustrates this issue quite well. In the top panel, there is indeed an analogy-like phrase ("tidying up"), but that analogy is not quite relevant to the task. And, at least in the verbal description, I don't see the correct solution. (the solution, I believe is that different shapes fall to the right or to the left in tetris-like fashion, depending on the border color).

I believe that listing every such issue will be counterproductive. I believe that the authors have an interesting idea, but before diving into subtle and challenging evaluations of analogical relevance, it's crucial to first provide simple experiments with human participants to demonstrate that the resulting tasks are human-solvable.

Additionally, a simple way to test the quality of the generated analogies/solutions is to give them to humans and to ask to apply them to the task. To show that the analogies/solutions are complete/sufficient, humans should be able to solve the test example without seeing the training arc demonstrations. To show that the analogies are at least helpful, one can compare human performance on these tasks with and without the analogies. This is similar to some of the experiments shown in the paper, but the crucial thing is that the focus should be on task solving performance (not analogy identification) and it should be, in my view, on human participants.

Overall, unfortunately, at present, I can not recommend this paper to acceptance. But I hope that the authors refine their approach and resubmit the paper in the future.

**Questions:**

What was the reason for selecting specifically 3 experts and 12 problems?

In the appendix D, it's stated that the experts were shown 12 tasks. But in the provided prompt given to the experts, it's said that the number of tasks is 13. What is the cause for this discrepancy?

In 4.1. (i), there's a mention of iterative refinement in task selection. How were these tasks selected and what was involved in this refinement process?

---

> ### Author Response · Authors · 2025-11-23
>
> We appreciate your comments and suggestions.
>
> ---
>
> ## [Q1] What was the reason for selecting specifically 3 experts and 12 problems?
>
> We fully accept the reviewer’s critique that utilizing only 3 experts and 12 tasks is insufficient for generalization. Also, we agree with the reviewer's point that validating human solvability, which involves verifying that GIFARC tasks are indeed solvable and of high quality, is essential to prove the dataset's validity.
>
> To address this, we plan to recruit 10 experts to conduct a rigorous solvability check. Crucially, to ensure the results are generalizable and representative of the entire dataset, we will select test problems using stratified sampling based on the task categories and distribution outlined in Appendix A. Furthermore, to objectively assess dataset quality beyond human evaluation, we will introduce a new quantitative method to measure how accurately the analogies extracted from GIFs are grounded in the generated GIFARC tasks.
>
> Even when some analogies appear incomplete or even unsolvable from a human perspective, they may still serve as useful synthetic supervision signals by guiding the model toward an analogical reasoning approach rather than a purely pattern-matching strategy. In future work, we plan to systematically study this effect by training models on controlled variants of the dataset that differ along two axes: analogy correctness (correct, partially correct, and misleading analogies) and task solvability (tasks that human experts judge as solvable versus unsolvable). We will then compare these models in terms of both performance and their solution behavior.
>
> ---
>
> ## [Q2] In the appendix D, it's stated that the experts were shown 12 tasks. But in the provided prompt given to the experts, it's said that the number of tasks is 13. What is the cause for this discrepancy?
>
> The mention of '13' in the guidelines was a typographical error that occurred during the manuscript editing process. As correctly indicated in the Section 4.2, the actual test was strictly conducted with 12 tasks. We apologize for this confusion and will correct it in the revision.
>
> ---
>
> ## [Q3] In 4.1. (i), there's a mention of iterative refinement in task selection. How were these tasks selected and what was involved in this refinement process?
>
> The multi-turn refinement mentioned in Section 4.1 (i) refers to the process of preparing 15 "full description" samples. Regarding representativeness, these tasks were not intended as a statistically random subset of GIFARC. Instead, they were selected primarily from the perspective of conceptual clarity and visual salience. We prioritized samples where the core analogical concepts, such as "gravity" or "wave", were unambiguously manifested in both the grid transformation and the textual description. This selection was intentional, as providing the model with the most distinct and instructive examples is considered more critical for effective in-context learning than adhering to a random statistical distribution.
>
> The refinement process involved providing the LLM with these distinct descriptions and iteratively prompting it to strip away non-essential details, ensuring the text focused solely on the clear analogical mapping. Subsequently, we conducted a manual review to verify this alignment. Thus, these 15 samples represent the "clearest reasoning templates" within the dataset, serving as strong anchors to guide the model's approach to more complex or subtle tasks in the test set. We will incorporate these clarifications into the revised manuscript.
>
> While our selection was intentional as described above, we acknowledge the reviewer's valid concern regarding representativeness. Therefore, in the revised submission, we will expand the evaluation to 100-200 tasks through stratified sampling across the 20 task categories and code complexity levels documented in Appendix A, ensuring representativeness and sufficient sample size.

---

### Meta-Review · Area_Chair_DJKY · 2026-01-07

**Summary:**

The paper received feedback from four reviewers, who reached a strong consensus regarding the work. All reviewers acknowledged that the core concept—leveraging GIFs to extract visual analogies for generating ARC (Abstract Reasoning Corpus) tasks—is both novel and highly promising. The motivation to bridge the gap between dynamic visual intuition and static logical reasoning was well-received.


However, significant shortcomings remain regarding the execution and validation of this idea. The reviewers collectively identified three critical areas where the paper falls short:
（1）Task Solvability & Explainability: There is insufficient evidence to prove that the generated tasks are solvable by humans or that the underlying logic is explainable.
（2）Data Quality Reliability: The mapping between the source GIFs and the resulting grid tasks was often found to be tenuous, raising doubts about the quality of the synthetic dataset.
（3）Evaluation Validity: The methodology, particularly the reliance on LLM-based self-evaluation, was deemed circular and lacked objective rigor.


While the authors provided detailed rebuttals and attempted to offer more reasonable justifications during the discussion phase, their responses often remained at a conceptual or narrative level. The revision lacked concrete experimental evidence and robust quantitative data analysis sufficient to quell the reviewers' doubts (e.g., comprehensive human performance baselines or external validation metrics). Consequently, the work currently lacks the empirical rigor required for acceptance.

Final Decision: The Area Chair (AC) concurs with the reviewers' detailed assessment. Although the paper introduces an interesting direction for the ARC domain, the unresolved technical and methodological issues still raise a decision of Rejection. The Area Chair strongly encourages the authors to carefully address the specific feedback provided by the reviewers. With these substantial improvements, the AC believes this work has the potential to be a strong candidate for future top-tier conferences.

**Reviewer Concerns:**

The primary concerns raised by the reviewers fall into two broad categories. The first category pertains to technical implementation details and clarity. The manuscript contained several ambiguous descriptions, prompting reviewers to pose numerous clarification questions to verify specific mechanisms and technical details. The second category concerns the scientific validity and rigor of the work. These issues specifically revolve around task solvability and explainability, data quality and reliability, and evaluation validity.

While the authors provided comprehensive and reasonable responses to the first category, effectively resolving the technical ambiguities, they failed to offer substantial improvements or convincing evidence to address the fundamental flaws in the second category during the rebuttal.

**Reviewer Scores:**

After a careful examination of the reviews and the authors' rebuttal, the AC believes it is doubtful that reviewers will raise their scores, particularly to the extent of shifting a decision from Rejection to Acceptance.

The core reason for this assessment is that, while the authors attempted to provide more reasonable justifications for the critical issues raised, their responses largely remained at the level of conceptual discussion or narrative explanation. Concrete experimental verification and rigorous data analysis are still conspicuously absent, resulting in a continued lack of rigor.

Consequently, it is evident that the reviewers require stronger empirical evidence—specifically performing additional experiments and providing robust statistical data—to effectively address their concerns. As emphasized by multiple reviewers in their responses, the magnitude of the required work implies that the revised paper and re-reviewing will not be ready for ICLR this year.

---

### Decision · Program_Chairs · 2026-01-26

Reject